# Study on Sound-Insulation Performance of an Acoustic Metamaterial of Air-Permeable Multiple-Parallel-Connection Folding Chambers by Acoustic Finite Element Simulation

**DOI:** 10.3390/ma16124298

**Published:** 2023-06-09

**Authors:** Wenqiang Peng, Shaohua Bi, Xinmin Shen, Xiaocui Yang, Fei Yang, Enshuai Wang

**Affiliations:** 1College of Aerospace Science and Engineering, National University of Defense Technology, Changsha 410073, China; 2Field Engineering College, Army Engineering University of PLA, Nanjing 210007, China; 17337434454@163.com (S.B.); 19962061916@163.com (F.Y.); 3Engineering Training Center, Nanjing Vocational University of Industry Technology, Nanjing 210023, China; 2019101052@niit.edu.cn; 4MIIT Key Laboratory of Multifunctional Lightweight Materials and Structures (MLMS), Nanjing University of Aeronautics and Astronautics, Nanjing 210016, China; wangenshuai0823@126.com

**Keywords:** acoustic metamaterial, air-permeable multiple-parallel-connection folding chambers, Fano-like interference, acoustic finite element simulation, sound insulation performance, parametric analysis, sound transmission loss

## Abstract

In order to achieve a balance between sound insulation and ventilation, a novel acoustic metamaterial of air-permeable multiple-parallel-connection folding chambers was proposed in this study that was based on Fano-like interference, and its sound-insulation performance was investigated through acoustic finite element simulation. Each layer of the multiple-parallel-connection folding chambers consisted of a square front panel with many apertures and a corresponding chamber with many cavities, which were able to extend both in the thickness direction and in the plane direction. Parametric analysis was conducted for the number of layers *n_l_* and turns *n_t_*, the thickness of each layer *L*_2_, the inner side lengths of the helical chamber *a*_1_, and the interval *s* among the various cavities. With the parameters of *n_l_* = 10, *n_t_* = 1, *L*_2_ = 10 mm, *a*_1_ = 28 mm, and *s* = 1 mm, there were 21 sound-transmission-loss peaks in the frequency range 200–1600 Hz, and the sound-transmission loss reached 26.05 dB, 26.85 dB, 27.03 dB, and 33.6 dB at the low frequencies 468 Hz, 525 Hz, 560 Hz, and 580 Hz, respectively. Meanwhile, the corresponding open area for air passage reached 55.18%, which yielded a capacity for both efficient ventilation and high selective-sound-insulation performance.

## 1. Introduction

The noise pollution generated by urbanization and socioeconomic development not only threatens human health [1], but from it can arise also a series of social problems [2], especially in urban areas [3]. Yang et al. [4] evaluated traffic noise pollution based on noise maps for the Chancheng District in Foshan, China, which exhibited a high magnitude of noise and long-lasting noise pollution near streetfront buildings as well as in areas where quietness was required. The exposure to road-traffic air and noise pollution was estimated by Khan et al. [5], who aimed to exhibit the associated challenges, research gaps, and priorities. Gupta et al. [6] analyzed and summarized the adverse effects of indoor and outdoor environmental noise pollution on fetuses, infants, children, adolescents, and adults. Based on the observed equivalent noise data of 113 major Chinese cities, a Bayesian spatiotemporal hierarchy model was employed by Jin et al. [7] to investigate the spatiotemporal characteristics of urban noise pollution in China from 2007 to 2019. These research achievements indicated that noise-pollution prevention is essential for sustainable development.

Many sound-absorbing and insulation materials have been developed to reduce noise pollution, such as porous media [8], microperforated panels [9], acoustic metamaterials [10], composite structures [11], etc. Gradient-compressed porous metals were developed and optimized by Yang et al. [12], and the optimal average sound-absorption coefficient of 0.6033 in 100–6000 Hz was obtained with total thickness of 11 mm. Multilayer microperforated panel absorbers have been utilized to enlarge the sound-absorption bandwidth [13,14,15,16,17]. In order to overcome the shortcomings of single sound-absorbing materials and structures, composite sound-absorbing structures of porous materials and microperforated panels have been proposed [18,19,20,21]. In order to obtain better sound-absorption performance in the low-frequency range, many effective acoustic metamaterials have been developed, which can obtain excellent sound-absorption properties with a thickness of one in dozens of subwavelengths [22,23,24,25]. The proposed and developed sound absorbers promote noise-pollution prevention and improve the acoustic environment.

High-power equipment is one of the main sources generating annoying noise, and its noise reduction must take the heat radiation into account. The noise generated by large machines and equipment with periodic rotating or reciprocating excitations was investigated by Cong et al. [26], and it often contained low-frequency harmonic components which would be a challenge for some applications. It was pointed out by Chiu [27] that noise control was important and essential in an enclosed machine room where the noise level was restricted by the Occupational Safety and Health Act. A membrane acoustic metamaterial which was composed of a prestressed membrane and attached mass was proposed by Xu et al. [28], and it was expected to be applied in noise control, such as noise reduction for mechanical equipment and architectural acoustics. Song et al. [29] investigated the noise of milling machines and considered its characteristics for application to an efficient soundproofing material for noise reduction, which resulted in the application of Thinsulator and a triple soundproofing mat (butyl 100% + aluminum + sound-insulating material), which was effective in noise reduction for milling machines. However, the ventilation performances of these sound absorbers were unsatisfactory, which indicates that a novel acoustic metamaterial with high open area for air passage is desired. Meanwhile, in most instances, the noises generated during rotational motion are not continuous, and their distributions are at several frequency points in a certain frequency range. Therefore, the noise control for this circumstance requires the sound-insulation properties of the utilized material or structure to be accurate at these frequency points, rather than broadband noise reduction, which is favorable to decreasing the total thickness of the sound insulator, improving the sound-insulation efficiency, and reducing the occupied space.

Based on Fano-like interference, a deep-subwavelength acoustic-metasurface unit cell comprising nearly 60% open area for air passage was proposed by Ghaffarivardavagh et al. [30], which could serve as a high-performance selective sound silencer and which might find utility in the applications where highly efficient, air-permeable sound silencers are required, such as for smart sound barriers, fan- or engine-noise reduction, etc. It can realize high sound-insulation performance by the interference of the sound waves penetrating the central area with those passing through the surrounding spiral chambers, yielding a novel acoustic metamaterial to obtain sound insulation and ventilation simultaneously. This was pioneering of Ghaffarivardavagh et al., and provided effective guidance for the development of more practical sound insulators. Using this method, some novel acoustic metamaterials have been developed, such as a ventilated acoustic meta-barrier based on layered Helmholtz resonators [31], a three-dimensional reticular structure made with spheres [32], a ventilation barrier with space-coiling channels [33], a subwavelength acoustic metamaterial based on labyrinthine structures [34], an arch-like labyrinthine acoustic metasurface [35], a ventilated soundproof acoustic metamaterial consisting of resonant cavities arranged around a central air passage [36], nonlocal ventilating metasurfaces [37], etc. However, the shapes of these air-permeable acoustic metamaterials were complex, which meant that corresponding fabrications were difficult to realize and that their actual applications were limited. Meanwhile, the number of sound-transmission-loss peaks was limited, being generated from the ultra-open acoustic-metamaterial silencer, and this was a major shortcoming of this groundbreaking sound insulator. Moreover, numerical modeling methods could improve the research efficiency and reduce the research cost simultaneously, for example, finite element simulations [38,39,40,41,42,43], the finite difference method [44], the Bezier multi-step method [45], the differential quadrature method [46], etc. 

Therefore, an acoustic metamaterial of air-permeable multiple-parallel-connection folding chambers has been proposed in this study, which aimed to achieve excellent sound-insulation performance with a simple structure. Acoustic finite element simulation models of an ultra-open acoustic-metamaterial silencer, multiple-parallel-connection helical chambers, and multiple-parallel-connection folding chambers were constructed successively, and their influencing factors were analyzed as well. Meanwhile, the sound-insulation mechanisms of these acoustic metamaterials were revealed and exhibited in an acoustic finite element simulation [38,39,40,41,42,43]. The proposed air-permeable multiple-parallel-connection folding chambers can balance the sound-insulation performance and the structural complexity, which is favorable for promoting its practical application.

## 2. Materials and Methods

Based on Fano-like interference, the acoustic metamaterial of air-permeable multiple-parallel-connection folding chambers was proposed and developed in this study, which evolved from the ultra-open acoustic-metamaterial silencer [30]. Therefore, acoustic finite element simulation models for an ultra-open acoustic-metamaterial silencer, multiple-parallel-connection helical chambers, and multiple-parallel-connection folding chambers were constructed, which provided the foundation to investigate the influences of structural parameters on the sound-insulation performance.

### 2.1. Ultra-Open Acoustic-Metamaterial Silencer

Based on the pressure-acoustics module in the COMSOL Multiphysics v5.5 software, the acoustic finite element simulation model to investigate the sound-insulation performance of an ultra-open acoustic-metamaterial silencer was built, which consisted of inlet, incident channel, ultra-open acoustic-metamaterial silencer, transmission channel, and the outlet, as shown in Figure 1a.

The investigated frequency range was 200–1600 Hz in this study, which took into account the normal frequency distribution of main noise generated by the rotational motion of common equipment. Normally speaking, the simulation accuracy will be higher when the finite element size is smaller, and the appropriate element size is 1/50 of the minimum wavelength (1/50 × 340/1600 × 1000 ≈ 4.25 mm) for normal structures, and it should be 1/1000 of the minimum wavelength (1/1000 × 340/1600 × 1000 ≈ 0.2 mm) for complex structures. The grid model is shown in Figure 1b, and the maximum and minimum element sizes were set as 5 mm and 1 mm, respectively, and the utilized element type was regular tetrahedral mesh in this research. Meanwhile, in order to improve the accuracy of the investigated acoustic metamaterial, the minimum element size was set as 0.025 mm for the domain of the ultra-open acoustic-metamaterial silencer. The main structural parameters of the ultra-open acoustic-metamaterial silencer included the total thickness *T*_a_, the outer and the inner radius of the spiral chamber *r*_2_ and *r*_1_, the occupied sectional angle and helix angle of each spiral chamber *k*_0_ and *θ*, and the thickness of the side walls *t*_0_, as shown in Figure 1c–e. Meanwhile, both the thickness of incident channel *T*_i_ and that of the transmission channel *T*_t_ were set as 2 × *T*_a_. The open area for air passage *μ*_open_ was defined as the ratio between the area of the central straight hole and that of whole ultra-open acoustic-metamaterial silencer, as shown in Equation (1).
(1)$$\mu_{open}=\frac{\pi {\left(r_{1}-t_{0}\right)}^{2}}{\pi {\left(r_{2}+t_{0}\right)}^{2}}={\left(\frac{r_{1}-t_{0}}{r_{2}+t_{0}}\right)}^{2}$$

An incident sound wave with the amplitude of 1 Pa and the phase of 0 rad was introduced at the inlet, and it penetrated the incident channel and reached the ultra-open acoustic-metamaterial silencer. The sound wave passed the central straight hole and threaded the 6 spiral chambers and realized the interference phenomenon in the transmission channel, which could achieve excellent sound-insulation performance at some frequency point. That is the major sound-insulation mechanism of the ultra-open acoustic-metamaterial silencer [30]. The influences of the structural parameters of the total thickness *T*_a_, the open area for air passage *μ*_open_, the occupied sectional angle *k*_0_, and the helix angle of each spiral chamber *θ* were analyzed one by one, which aimed to improve the sound-insulation properties. The investigated parameters and their values are summarized in Table 1. The selected and default values investigated in this study considered practical application scenarios and the actual available range simultaneously. For example, the total thickness *T*_a_ should be neither too big nor too small, because the occupied space would be too large when *T*_a_ is too big and the sound-insulation performance would be too weak when the *T*_a_ is too small. For another example, the occupied sectional angle *k*_0_ should be no more than 60 deg (360/6 = 60), otherwise the 6 spiral chambers would interfere. Moreover, the objective of the research on the impacts of these parameters was to show the evolution of sound-insulation performance with variable parameters, which could lead to the acoustic metamaterials proposed in this research.

Meanwhile, the other parameters *r*_2_ and *t*_0_ were set as 96 mm and 2 mm, respectively, which aimed to make the size of the ultra-open acoustic-metamaterial silencer to be Ф100 mm. For each investigated parameter, 5 values were selected, along with the default values for the other parameters, which could exhibit the sound-insulation performance of the ultra-open acoustic-metamaterial silencer and provide guidance to develop the novel acoustic metamaterials to achieve the balance between sound insulation and ventilation.

It had been demonstrated that the ultra-open acoustic-metamaterial silencer could only yield a small number of sound-transmission-loss peaks, because the lengths of the 6 spiral chambers were equal. Relative to the sound wave in the central straight hole, the phase changes of the sound wave in the 6 spiral chambers were the same, which meant that the generated number of sound-transmission-loss peaks was limited. Although the amplitudes of the sound-transmission-loss peaks were large, the limited number seriously restricted its practical application value. Meanwhile, the shape of the ultra-open acoustic-metamaterial silencer was complex, because there were many free-form curves and three-dimensional curves, which meant that the fabrication of actual samples was difficult to realize, especially since the complex shapes had some internally complex structures. The normal processing methods for material removal could not be utilized in the fabrication of the ultra-open acoustic-metamaterial silencer, because the cutting tools were unable to enter its internal areas for effective material machining. Although the additive manufacturing method could fabricate the acoustic metamaterials with the complex shapes, this would be difficult to use in mass production. Moreover, the complex shapes of the ultra-open acoustic-metamaterial silencer made the mold-manufacturing method difficult to realize with a low production cost and high yield rate. Therefore, in order to reduce the manufacturing difficulty effectively, the acoustic material of the multiple-parallel-connection helical chambers was proposed and developed in this study.

### 2.2. Multiple-Parallel-Connection Helical Chambers

Similarly, the acoustic finite element simulation model to investigate the sound-insulation performance of the multiple-parallel-connection helical chambers was constructed, as shown in Figure 2a. The ultra-open acoustic-metamaterial silencer was replaced by multiple-parallel-connection helical chambers as seen in the geometry model in Figure 1a, and the set of element sizes and the selection of element types in Figure 2b are the same as those in Figure 1b.

The main parameters of the multiple-parallel-connection helical chambers included the outer and inner side lengths of the helical chambers *a*_2_ and *a*_1_, the side length of the square aperture *a*_3_, the thickness of each layer *L*_2_, the length of each straight chamber *L*_3_, and the thickness of the side walls *t*_0_, as shown in Figure 2c. It can be observed from Figure 2d that the initial spiral chamber in Figure 1e was replaced by the helical chamber in Figure 2d, which consists of several square apertures on each layer, straight chambers in each layer, and twisted chambers at the corner positions. Meanwhile, the open area for air passage *μ*_open_ can be derived by Equation (2).
(2)$$\mu_{open}=\frac{{\left(2a_{1}-2t_{0}\right)}^{2}}{{\left({2a}_{2}+2t_{0}\right)}^{2}}={\left(\frac{a_{1}-t_{0}}{a_{2}+t_{0}}\right)}^{2}$$

The 6 spiral chambers of the ultra-open acoustic-metamaterial silencer in Figure 1d were replaced by 4 helical chambers for the multiple-parallel-connection helical chambers in Figure 2c, which reduced the fabrication difficulty. It can be judged from Figure 2c,d that there were no complex shapes needed for the helical chambers, and that they could be divided into several layers. Each layer consisted of the front panel and corresponding cavity. There were several apertures on each front panel, and there were some channels on each cavity. Thus, all of the multiple-parallel-connection helical chambers could be assembled with several front panels and cavities layer-by-layer, which significantly reduces the fabrication cost and improves the production efficiency relative to that of the initial ultra-open acoustic-metamaterial silencer. However, because the lengths of the 4 helical chambers were equal, they could only achieve a small number of sound-transmission-loss peaks. Thus, in order to increase the number of sound-transmission-loss peaks, multiple-parallel-connection folding chambers were also developed, which had multiple folding chambers with different lengths. Meanwhile, as a transitional intermediate objective, the sound-insulation characteristics of the multiple-parallel-connection helical chambers were investigated simply by changing their total thickness. The selected structural parameters were as follows: the outer side length of helical chamber *a*_2_, 33 mm; the inner side length of helical chamber *a*_1_, 18 mm; the side length of the square aperture *a*_3_, 15 mm; the thickness of each layer *L*_2_, 10 mm; the length of each straight chamber *L*_3_, 49 mm; and the thickness of the side walls *t*_0_, 2 mm. The investigated numbers of layers were 5, 8, and 12, which meant that the total thicknesses of the acoustic metamaterial of the multiple-parallel-connection helical chambers were 62 mm (5 × *L*_2_ + 6 × *t*_0_), 98 mm (8 × *L*_2_ + 9 × *t*_0_), and 146 mm (12 × *L*_2_ + 13 × *t*_0_), respectively.

### 2.3. Multiple-Parallel-Connection Folding Chambers

Likewise, the acoustic finite element simulation model to investigate the sound-insulation performance of the multiple-parallel-connection folding chambers was constructed, as shown in Figure 3a. The ultra-open acoustic-metamaterial silencer was replaced by the multiple-parallel-connection folding chambers as seen in the geometry model in Figure 1a, and the set of element sizes and the selection of element types in Figure 3b are the same as those in Figure 1b.

Some of the same parameters were used for the various folding chambers in the multiple-parallel-connection folding chambers, including the outer and inner side lengths of the helical chambers *a*_2_ and *a*_1_, the side length of the square aperture *a*_3_, the thickness of each layer *L*_2_, and the thickness of the side walls *t*_0_, as shown in Figure 3c,d. Different from the helical chambers with the same length *L*_3_ in Figure 2c, the 8 folding chambers in Figure 3c had various lengths, which could be labelled as *L*_3*i*_ (I = 1, 2, …, 8). Meanwhile, the open area for air passage *μ*_open_ for the multiple-parallel-connection folding chambers could be derived by Equation (2) as well.

The 4 helical chambers with the same length in the multiple-parallel-connection helical chambers in Figure 2c were further replaced by 8 folding chambers with various lengths for the multiple-parallel-connection folding chambers in Figure 3c, which was favorable for achieving an additional reduction in fabrication difficulty. It can be seen from Figure 3c that the multiple-parallel-connection folding chambers were periodic with the change cycle of 2 layers, and there were no twisted chambers. Thus, the whole acoustic metamaterial of the multiple-parallel-connection folding chambers could be divided into 2 kinds of front panels and 2 kinds of corresponding cavities, which made large-scale assembly-line production possible. The front panels and cavities could achieve rapid mass production by mold manufacturing, and the expected sound-insulation property would be obtained by a combination of a certain number of front panels and cavities. Moreover, the lengths of the 8 folding chambers were different, which meant that the sound wave passed the central straight hole and threaded the 8 folding chambers and realized the interference phenomenon with the various frequency points in the transmission channel, and this was favorable for generating a higher number of sound-transmission-loss peaks. Therefore, relative to the ultra-open acoustic-metamaterial silencer and the multiple-parallel-connection helical chambers, the multiple-parallel-connection folding chambers had the advantage of lower fabrication cost and better sound-insulation performance, which made this one of the best potential candidates to realize the balance between sound insulation and ventilation. The influences of the structural parameters of the numbers of layers *n_l_*, the thickness of each layer *L*_2_, the inner side lengths of the helical chambers *a*_1_, and the interval among the various folding chambers *s* were analyzed one by one, as shown in Table 2, which process aimed to thoroughly exhibit the sound-insulation performance and effectively improve the sound-insulation properties. Meanwhile, the value of outer side length of helical chamber *a*_2_ was set as 33 mm, and the side length of the square aperture *a*_3_ was set equal to (*a*_2_ − *a*_1_)/2.

## 3. Results and Discussions

Based on the acoustic finite element simulation models constructed, the sound-insulation performances of the ultra-open acoustic-metamaterial silencer, the multiple-parallel-connection helical chambers, and the multiple-parallel-connection folding chambers were analyzed successively, and suitable application conditions and scenarios were summarized for these three novel acoustic metamaterials in the field of noise reduction.

### 3.1. Sound-Insulation Property of Ultra-Open Acoustic-Metamaterial Silencer

For the ultra-open acoustic-metamaterial silencer in Figure 1, the influences of its structural parameters on the corresponding sound-insulation performances were investigated one by one according to the selected parameters in Table 1, to exhibit its special sound-insulation characteristics.

#### 3.1.1. Total Thickness *T_a_*

The sound-insulation performances corresponding to the various total thicknesses *T*_a_ are shown in Figure 4. It can be observed that in the investigated frequency range 200–1600 Hz, the number of sound-transmission-loss peaks increased from two to five along with the increase in total thickness *T*_a_ from 50 mm to 130 mm, and the corresponding first sound-transmission peak shifted from 670 Hz to 242 Hz. It was found that the ultra-open acoustic-metamaterial silencer could achieve some sound-transmission peaks, but the realization of peak frequencies in the low-frequency range required a large total thickness, which meant that the occupied space would be large and the fabrication difficult.

For the ultra-open acoustic-metamaterial silencer with various total thicknesses *T*_a_, these sound-transmission-loss peaks *STL*_p_ and the corresponding frequency points *f*_p_ are summarized in Table 3. It can be observed that, along with the increase in total thickness, the sound-transmission-loss peaks *STL*_p_ decreased gradually and the peak frequency *f*_p_ shifted in the low-frequency direction simultaneously, which indicated that the addition of sound-insulation capacity by the increase in space mainly contributed to the generation of new sound-transmission-loss peaks.

#### 3.1.2. Open Area for Air Passage *μ*_open_

The sound-insulation performances corresponding to the various open areas for air passage *μ*_open_ are shown in Figure 5. When the *μ*_open_ was 0.5, the corresponding sound-insulation performance was the same as that in Figure 4c. It was observed that no matter how large the *μ*_open_ was, there were four sound-transmission-loss peaks in the frequency range of 200–1600 Hz, which indicated that the shift of sound-transmission-loss peaks was insignificant with the alteration of the open area for air passage. For the ultra-open acoustic-metamaterial silencer with various open areas for air passage *μ*_open_, these sound-transmission-loss peaks *STL*_p_ and the corresponding frequency points *f*_p_ are summarized in Table 4. It can be observed that the sound-transmission-loss peak *STL*_p_ decreased gradually and the corresponding frequency point *f*_p_ shifted in the low-frequency direction along with the increase in the open area for air passage *μ*_open_, which indicated that the sound insulation for low frequencies was more difficult relative to that for high frequencies, with similar sound-insulation capability for a certain total thickness of the ultra-open acoustic-metamaterial silencer. Thus, the adjustment of *μ*_open_ could alter the interference frequency, but the sound-transmission-loss peak changed accordingly.

#### 3.1.3. Occupied Sectional Angle *k*_0_

The sound-insulation performances corresponding to the various occupied sectional angles *k*_0_ are shown in Figure 6. When the *k*_0_ was 50 deg, the corresponding sound-insulation performance was the same as that in Figure 4c. There were four sound-transmission-loss peaks in the investigated frequency range 200–1600 Hz with the various values for *k*_0_, which were similar to those for the various open areas for air passage *μ*_open_.

For the ultra-open acoustic-metamaterial silencer with the various occupied sectional angles *k*_0_, these sound-transmission-loss peaks *STL*_p_ and corresponding frequency points *f*_p_ are summarized in Table 5.

It was observed that the interference frequency *f*_p_ shifted in the high-frequency direction gradually along with the increase in the occupied sectional angle *k*_0_, but the displacements were quite limited, being 11 Hz, 10 Hz, 26 Hz, and 28 Hz for the first, second, third, and fourth peaks, respectively. Meanwhile, it was found that there was no obvious pattern in the variation of the sound-transmission-loss peaks *STL*_p_ with the alteration of the occupied sectional angle *k*_0_, and this indicated that the *k*_0_ should be strictly selected for the desired sound-insulation performance.

#### 3.1.4. Helix Angle of Each Spiral Chamber *θ*

The sound-insulation performances corresponding to the various helix angles of each spiral chamber *θ* are shown in Figure 7. When the *θ* was 0.14 rad, the corresponding sound-insulation performance was the same as that in Figure 4c. These sound-transmission-loss peaks *STL*_p_ and the corresponding frequency points *f*_p_ are summarized in Table 6.

It was observed that the interference frequency *f*_p_ shifted in the high-frequency direction gradually and significantly along with the increase in the helix angle of each spiral chamber *θ*, because the length of each spiral decreased gradually with the increase in *θ* for the given total thickness. The displacements were 105 Hz, 190 Hz, 299 Hz, and 400 Hz for the first, second, third, and fourth peaks, respectively. The appearance was obviously different from that for the various occupied sectional angles *k*_0_, because the *k*_0_ only determined the incident range of the acoustic wave, while the *θ* could determine the length of the spiral chamber for the passage of the acoustic wave. On the other hand, there was no obvious pattern in the variation of the sound-transmission-loss peaks *STL*_p_ with the alteration of the helix angle of each spiral chamber *θ*, which was similar to the appearance with the occupied sectional angles *k*_0_. The undulation in the values of the sound-transmission-loss peaks *STL*_p_ indicated that the parameter *θ* should be carefully selected together with the total thickness *T*_a_, the open area for air passage *μ*_open_, and the occupied sectional angle *k*_0_. All the sound-transmission-loss peaks *STL*_p_ were above 26 dB, which indicated that the ultra-open acoustic-metamaterial silencer achieved fine sound-insulation properties.

#### 3.1.5. Summary and Analysis

It was found that the major influencing factor for the sound-insulation property of ultra-open acoustic-metamaterial silencer was total thickness *T*_a_, and the helix angle of each spiral chamber *θ* and the occupied sectional angle *k*_0_ could be adjusted for the expected sound-insulation performance. By changing the shape from round to square, decreasing the number of chambers from six to four, and selecting the parameters as *θ* = 0 and *k*_0_ = 90, the initial ultra-open acoustic-metamaterial silencer evolved into multiple-parallel-connection helical chambers.

### 3.2. Sound-Insulation Property of Multiple-Parallel-Connection Helical Chambers

Regarding the multiple-parallel-connection helical chambers in Figure 2, the influence of the number of layers on the sound-insulation performance was investigated in this study to exhibit the sound-insulation characteristics and the corresponding boundedness in the actual practical application for noise reduction. The acoustic finite element simulation model of the multiple-parallel-connection helical chambers with various numbers of layers is shown in Figure 8, and the corresponding sound-insulation performances are shown in Figure 9. Meanwhile, the sound-insulation properties of the multiple-parallel-connection helical chambers with various numbers of layers *n_l_* are summarized in Table 7. It was observed that, along with the increase in numbers of layers, the interference frequency *f*_p_ clearly shifted in the low-frequency direction, but the number of sound-transmission-loss peaks only increased from two to three with the increase in the total thickness from 62 mm (5 × *L*_2_ + 6 × *t*_0_) to 146 mm (12 × *L*_2_ + 13 × *t*_0_). It could be judged from Figure 4e that the ultra-open acoustic-metamaterial silencer achieved five sound-transmission-loss peaks with a total thickness of 130 mm. Therefore, the capability of the multiple-parallel-connection helical chambers was weaker than that of the ultra-open acoustic-metamaterial silencer in generating sound-transmission-loss peaks. However, it was observed that the sound-insulation performances of those two peaks were above 20 dB, which indicated that the multiple-parallel-connection helical chambers would be favorable for generating a limited frequency band with high sound-insulation properties.

### 3.3. Sound-Insulation Property of Multiple-Parallel-Connection Folding Chambers

As for the multiple-parallel-connection folding chambers in Figure 3, the influences of the structural parameters on the corresponding sound-insulation performances were investigated one by one according to the selected parameters in Table 3, which consisted of the number of layers *n_l_*, the thickness of each layer *L_2_*, the inner side lengths of the helical chambers *a*_1_, and the interval among the various folding chambers *s*. By comparing its sound-insulation performances with the other two acoustic metamaterials, the comparative advantages and practical values were exhibited.

#### 3.3.1. Number of Layers *n_l_*

The acoustic finite element simulation model of the multiple-parallel-connection folding chambers with various numbers of layers is shown in Figure 10, and the corresponding sound-insulation performances are shown in Figure 11.

It was observed that, along with the numbers of layers *n_l_*, the interference frequency *f*_p_ shifted in the low-frequency direction significantly. It could be judged from the increase in the number of sound-transmission-loss peaks in the investigated frequency range. There were eight sound-insulation-loss peaks in Figure 11a because there were eight folding chambers with different lengths in Figure 10. Along with the increase in the number of layers, the second and third groups of sound-transmission-loss peaks appeared successively, as shown in Figure 11d.

#### 3.3.2. Thickness of Each Layer *L*_2_

Except for a thickness of each layer *L*_2_ = 10 mm, shown in Figure 10a, the acoustic finite element simulation model of the multiple-parallel-connection folding chambers with the other various thicknesses of each layer is shown in Figure 12, and the corresponding sound-insulation performances are shown in Figure 13.

It can be observed that the number of sound-transmission-loss peaks was kept at eight for the four conditions, and the interference frequency *f*_p_ shifted in the low-frequency direction significantly along with the increase in thickness of each layer *L*_2_, because the volume of each folding chamber increased gradually and this was favorable to the generation of sound-transmission-loss peaks at lower frequency points. The number of sound-transmission-loss peaks rose markedly relative to the ultra-open acoustic-metamaterial silencer and the multiple-parallel-connection helical chambers.

The sound-insulation performance of the multiple-parallel-connection folding chambers with various thicknesses of each layer *L*_2_ is summarized in Table 8.

It was found that the sound-insulation range (from the first peak to the eighth peak) was improved from 1258–1574 Hz to 993–1196 Hz with the increase in thickness of each layer *L*_2_ from 6 mm to 14 mm. Meanwhile, the average sound-transmission losses of the eight peaks were 29.75 dB, 33.01 dB, 29.36 dB, and 33.12 dB, respectively, with the various thicknesses of each layer *L*_2_ from 6 mm to 14 mm, which indicated that the corresponding sound-insulation property had not obviously been weakened with the frequency range shifting in the low-frequency direction, because the total sound-insulation capability rose along with the increase in *L*_2_.

#### 3.3.3. Inner Side Lengths of Helical Chamber *a*_1_

Except for an inner side length of the helical chamber *a*_1_ = 28 mm, shown in Figure 10a, an acoustic finite element simulation model of the multiple-parallel-connection folding chambers with the other various inner side lengths of helical chambers is shown in Figure 14, and the corresponding sound-insulation performances are shown in Figure 15.

It could be derived that the open area for air passage *μ*_open_ increased from 39.51% to 73.47% with the increase in the side length of the helical chamber *a*_1_ from 24 mm to 32 mm, and the numbers of sound-transmission-loss peaks increased accordingly as well.

#### 3.3.4. Interval among Various Folding Chambers s

Except for the interval among the various folding chambers *s* = 1 mm, shown in Figure 10a, an acoustic finite element simulation model of the multiple-parallel-connection folding chambers with the various other intervals among the different folding chambers is shown in Figure 16, and the corresponding sound-insulation performances are shown in Figure 17.

The length *L*_3*i*_ in the multiple-parallel-connection folding chambers was equal to 28.5 mm with *s* = 0 mm, and it was from 25 mm to 32 mm with *s* = 1 mm. Meanwhile, the length *L*_3*i*_ was from 17.5 mm to 38.5 mm with *s* = 3 mm. It can be seen from Figure 17a that there was only one sound-transmission-loss peak in the investigated frequency range 200–1600 Hz when the length of each folding chamber was equal to a low value, and the other sound-transmission-loss peaks exceeded this frequency range. In contrast, there were seven sound-transmission-loss peaks in 200–1600 Hz when the interval *s* was 3 mm, as shown in Figure 17b, and the first interference frequency reached 944 Hz; relative to the sound-insulation performance with *s* = 1 mm and shown in Figure 11a, the distribution of these sound-transmission-loss peaks shown in Figure 17b was more dispersed, because the values of the length range in the latter were larger than those in the former. Moreover, the average sound-transmission loss for these interference frequencies was 35.68 dB with *s* = 3 mm, which was larger than the 29.36 dB with *s* = 1 mm. The major reason for this might be that the disturbance among these folding chambers was weakened with a larger *s*. 

#### 3.3.5. Number of Turns *n_t_*

For all the above acoustic finite element simulation models, there was only one turn, whether it was the ultra-open acoustic-metamaterial silencer, the multiple-parallel-connection helical chambers, or the multiple-parallel-connection folding chambers. An acoustic finite element simulation model of the multiple-parallel-connection folding chambers with the turns *n_t_* = 2 is shown in Figure 18a, and the corresponding sound-insulation performance is shown in Figure 18b.

The utilized parameters were as follows: the numbers of layers, *n_l_* = 10; the thickness of each layer, *L*_2_ = 10 mm; the interval among the various folding chambers, *s* = 1 mm; the inner side lengths of the helical chambers for the two turns, *a*_11_ = 28 mm and *a*_12_ = 23 mm. Relative to the sound-insulation performance for one turn with the same other parameters shown in Figure 11d, some secondary sound-transmission-loss peaks were generated, but the values for these peaks were small. Therefore, within a limited range, the number of sound-transmission-loss peaks could be improved by adding new turns. Meanwhile, it can be seen by comparing Figure 11d and Figure 18b that the interference frequencies for the outside turn were almost unaffected, which meant that the generation of each sound-transmission-loss peak was mutually independent between the different turns.

#### 3.3.6. Sound-Insulation Mechanism

The distributions of sound pressures at the four interference frequencies *f*_p_ for the ultra-open acoustic-metamaterial silencer with the default parameter values are shown in Figure 19. It was found the phase differences were π, 2π, 3π, and 4π, respectively, which were consistent with the phenomenon exhibited in the literature [30].

The distributions of sound pressures at the interference frequencies *f*_p_ for the multiple-parallel-connection helical chambers when *n_l_* were 5 and 8 are shown in Figure 20.

It was found that the first sound-transmission-loss peak was generated with a phase difference of π and the second peak was generated with a phase difference of 2π, which values were consistent with those for the ultra-open acoustic-metamaterial silencer.

The distributions of sound pressures at the eight interference frequencies *f*_p_ for the multiple-parallel-connection folding chambers with the default parameter values are shown in Figure 21.

It was observed that each interference frequency corresponded to one folding chamber, and the value for *f*_p_ rose gradually along with the decrease in the length of the folding chamber. For each chamber, the first sound-transmission-loss peak was generated with the phase difference of π. Along with the increase in the number of layers *n_l_*, the subsequent series of sound-transmission-loss peaks were generated gradually. Taking the condition of *n_l_* = 10 in Figure 11d for an example, the second series of sound-transmission-loss peaks had been completely generated and the third series arose progressively.

### 3.4. Reliability Analysis

In this study, the acoustic finite element simulation method was selected to investigate the sound-insulation performance of three proposed acoustic metamaterials, and its reliability was further analyzed and discussed.

Firstly, the acoustic finite element simulation had been proven to be a reliable and effective method to investigate the sound-absorption performance and sound-insulation properties of acoustic metamaterials in many studies [47,48,49,50,51]. It can be judged from these studies [47,48,49,50,51] that the simulation results have been consistent with the experimental data when selecting the appropriate parameters in the simulation model. Qu et al. [48] investigated the normal incidence of sound-transmission loss with lightweight composite partitions using finite element simulations as well as experiments conducted in a standing wave tube, and it was proven that the simulation and experimental results were closely aligned. Acoustic finite element numerical simulation analysis of the sound-insulation-hood model was carried out by Wu et al. [49] using the acoustic software LMS Virtual Lab Acoustics v13.3, in which the simulation results were compared with experimental data to verify the correctness of the model, and the theoretical results showed a good agreement with experiment data. By combining the finite element model simulation and an impedance tube test, the effectiveness of membrane-type acoustic metamaterials with petal-like rings and the correctness of the numerical simulation were validated by Huang et al. [51]. Thus, the acoustic finite element simulation method was used to investigate the sound-insulation performance of the ultra-open acoustic-metamaterial silencer, the multiple-parallel-connection helical chambers, and the multiple-parallel-connection folding chambers in this research.

Secondly, the simulation results for the ultra-open acoustic-metamaterial silencer were validated with the experimental results reported in the literature [30]. In the literature [30], the acoustic metamaterial achieved a sound-transmission-loss peak at 460 Hz with a sound-transmission loss of 13.6 dB. The parameters in the literature [30] were as follows: the total thickness *T*_a_ = 52 mm; the inner radius of the chamber *r*_1_ = 51 mm; the outer radius of the chamber *r*_2_ = 70 mm; and the helix angle of each spiral chamber *θ* = 8.2°. As a comparison, the finite element simulation model of the ultra-open acoustic-metamaterial silencer constructed in this research achieved a sound-transmission-loss peak at 464 Hz with a sound-transmission loss of 35.94 dB. The parameters used in our study were as follows: the total thickness *T*_a_ = 50 mm, the inner radius of the chamber *r*_1_ = 67.9 mm (corresponding to the open area for air passage *μ*_open_ = 0.5), the outer radius of the chamber *r*_2_ = 96 mm, and the helix angle of each spiral chamber *θ* = 0.14 rad (≈8.0°). There were two methods to derive the sound-transmission loss. One was based on the sound pressure, which was abbreviated as *STL*_sp_ and was utilized to show the sound-insulation performance in the literature [30]; the other was based on the sound energy, which was abbreviated as *STL*_se_ and was utilized to show the sound-insulation performance in this study, and there existed a relationship *STL*_se_ = 2 × *STL*_sp_. It was found that the sound-insulation performance in the simulation in this study was similar to the experimental results in the literature [30].

Thirdly, there have been many studies mainly investigating sound-insulation performance by numerical simulation without any experimental validations [52,53,54,55,56,57,58,59,60,61]. For example, a versatile and fully coupled technique based on the finite-element–boundary-element model was developed by Sahu et al. [52], and the contributions of the individual components in transmitted energy were identified by numerical simulations. It was presented by Fan et al. [53] that theoretical analysis and finite element simulation demonstrated that an acoustic metamaterial based on membrane-coated perforated plates can effectively block acoustic waves over a wide low-frequency band regardless of incident angles. In the literature [54], composite rubber reinforced with hollow glass microspheres was considered a promising composite material for noise reduction, and its sound-insulation mechanism was studied based using an acoustic finite element simulation to obtain the appropriate parameters with certain constraint conditions. Kim [55] conducted a numerical simulation of sound-transmission loss for a double-panel structure with sonic crystal composed of distributed local resonators, and the results showed that the sound insulation could be significantly improved in the low-frequency range while reducing the total mass without increasing the thickness. Maurin and Kwapisz [56] investigated the sound-transmission loss of composite barriers by means of 3D-structure acoustic finite element simulation analysis, and the impact of different model parameters on the sound-transmission loss was evaluated as well. Thus, it was supposed that the research results obtained by acoustic finite element simulation were also reliable when using the suitable simulation parameters in the simulation model.

Finally, it is more difficult to detect the sound-transmission loss for the sound-insulation performance than to detect the sound-absorption coefficient for the sound-absorption performance. In our previous research, an AWA6290T standing wave tube detector had been utilized to measure the actual sound-absorption coefficients of the materials or structures [62,63]. However, this equipment was only suitable to detect the sound-transmission loss of material with a thickness less than 10 mm, and it could not detect the sound-insulation performance for the three proposed acoustic metamaterials in this study. The experimental apparatus to detect the sound-transmission loss of these acoustic metamaterials was very complex and the required size of the samples was quite large [64]. That is why many literatures about the sound-insulation performance of acoustic metamaterials use acoustic finite element simulation instead of experimental measurement. Therefore, these were the four major reasons why the acoustic finite element simulation method was utilized to investigate the three acoustic metamaterials in this research, with their experimental validations to be conducted in the future.

## 4. Conclusions

A study on the sound-insulation performance of an acoustic metamaterial of air-permeable multiple-parallel-connection folding chambers was conducted in this research, which aimed to balance sound insulation and ventilation. An initial ultra-open acoustic-metamaterial silencer, transitional multiple-parallel-connection helical chambers, and the final multiple-parallel-connection folding chambers were investigated by acoustic finite element simulation, and the major achievements were as follows.

In order to achieve an excellent sound-insulation performance with a simple structure, the novel acoustic metamaterial of multiple-parallel-connection folding chambers was generated from the initial ultra-open acoustic-metamaterial silencer, which consisted of eight folding chambers. Through adjusting the values of the lengths of these eight folding chambers and regulating the other parameters, the multiple-parallel-connection folding chambers could achieve a balance between sound insulation and ventilation.Relative to the initial ultra-open acoustic-metamaterial silencer and the transitional multiple-parallel-connection helical chambers, the proposed novel multiple-parallel-connection folding chambers could obtain more sound-transmission-loss peaks. With the parameters of *n_l_* = 10, *n_t_* = 1, *L_2_* = 10 mm, *a*_1_ = 28 mm, and *s* = 1 mm, there were 21 sound-transmission-loss peaks in the investigated frequency range of 200–1600 Hz, and the sound-transmission loss reached 26.05 dB, 26.85 dB, 27.03 dB, and 33.6 dB at the low frequencies 468 Hz, 525 Hz, 560 Hz, and 580 Hz, respectively, which exhibited the excellent sound-insulation performance in the low-frequency region.The major influencing parameters were investigated successively, which included the number of layers *n_l_*, the thickness of each layer *L*_2_, the inner side lengths of the helical chamber *a*_1_, and the intervals among the various folding chambers *s*. Through selecting the suitable parameters, the expected sound-insulation performance could be obtained, which consisted of the desired sound-transmission-loss peaks and the interference frequencies, and the reason for these sound-insulation characteristics were analyzed as well.The sound-insulation mechanisms of these three acoustic metamaterials were revealed by the distributions of sound pressures at interference frequencies *f*_p_. It had been proven that the series of sound-transmission-loss peaks corresponded to the phase differences of *n*π, which were appropriate for both the ultra-open acoustic-metamaterial silencer and the multiple-parallel-connection helical chambers. For the multiple-parallel-connection folding chambers, each interference frequency corresponded to one folding chamber, and the value for *f*_p_ rose gradually along with the decrease in the length of the folding chamber.

The proposed multiple-parallel-connection folding chambers showed its potential application value for noise while meeting the requirement of ventilation simultaneously, which could provide effective guidance for developing other novel acoustic metamaterials and enrich the theory of progressive acoustic metamaterials.

## Figures and Tables

**Figure 1 materials-16-04298-f001:**
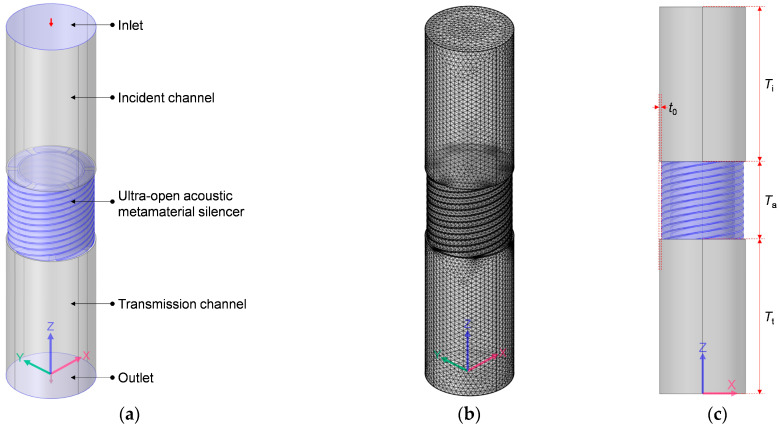
Acoustic finite element simulation model to investigate sound-insulation performance of the ultra-open acoustic-metamaterial silencer. (**a**) Geometry model; (**b**) grid model; (**c**) side view of geometry model; (**d**) geometry model of ultra-open acoustic-metamaterial silencer; (**e**) the single spiral chamber in the ultra-open acoustic-metamaterial silencer.

**Figure 2 materials-16-04298-f002:**
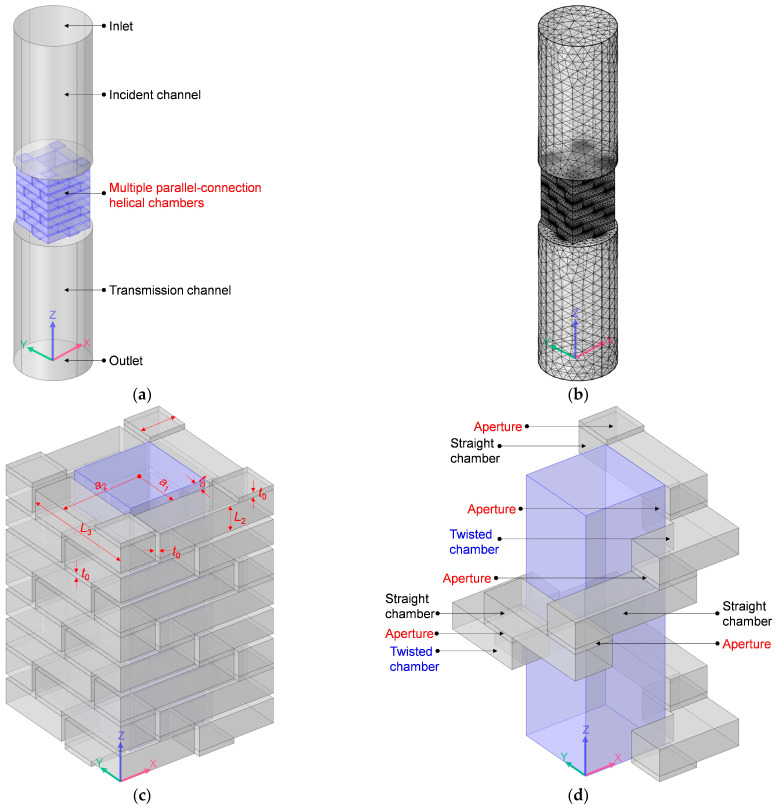
Acoustic finite element simulation model to investigate sound-insulation performance of the multiple-parallel-connection helical chambers. (**a**) Geometry model; (**b**) grid model; (**c**) geometry model of multiple-parallel-connection helical chambers; (**d**) single helical chamber in the multiple-parallel-connection helical chambers.

**Figure 3 materials-16-04298-f003:**
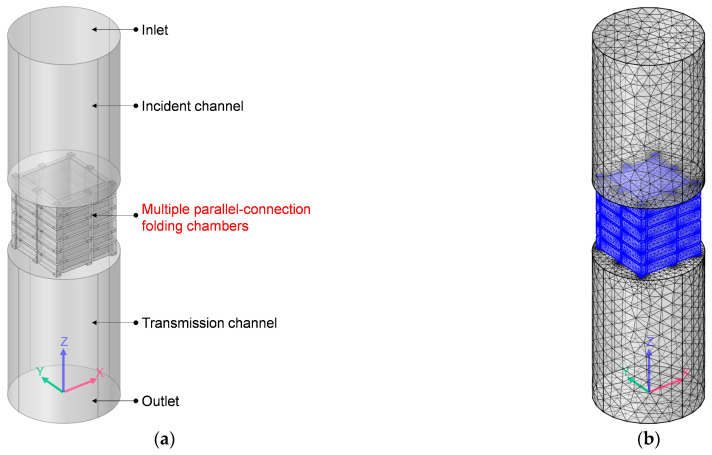
Acoustic finite element simulation model to investigate the sound-insulation performance of the multiple-parallel-connection folding chambers. (**a**) Geometry model; (**b**) grid model; (**c**) geometry model of multiple-parallel-connection folding chambers; (**d**) single folding chamber in the multiple-parallel-connection folding chambers.

**Figure 4 materials-16-04298-f004:**
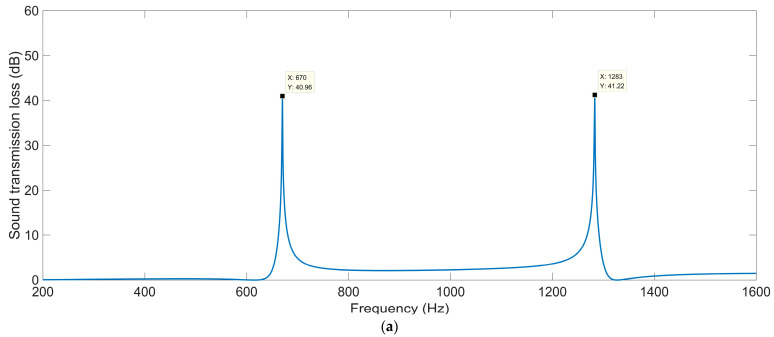
The sound-insulation performance of ultra-open acoustic-metamaterial silencer with various total thicknesses. (**a**) *T*_a_ = 50 mm; (**b**) *T*_a_ = 70 mm; (**c**) *T*_a_ = 90 mm; (**d**) *T*_a_ = 110 mm; (**e**) *T*_a_ = 130 mm.

**Figure 5 materials-16-04298-f005:**
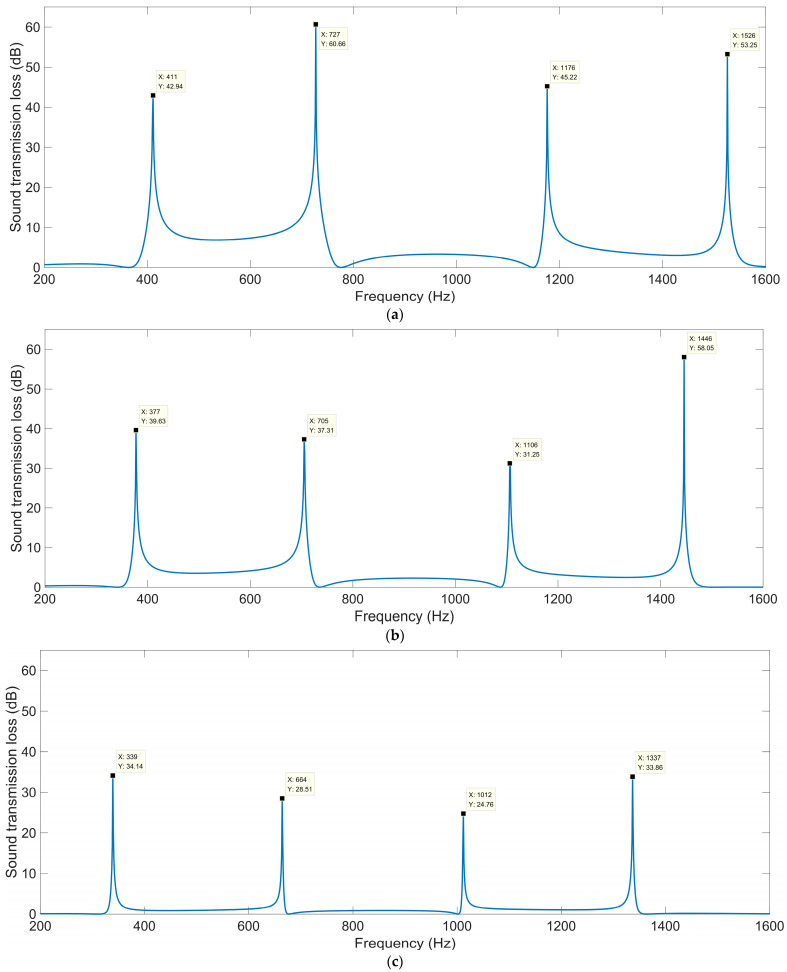
The sound-insulation performance of ultra-open acoustic-metamaterial silencer with the various open areas for air passage. (**a**) *μ*_open_ = 0.3; (**b**) *μ*_open_ = 0.4; (**c**) *μ*_open_ = 0.6; (**d**) *μ*_open_ = 0.7.

**Figure 6 materials-16-04298-f006:**
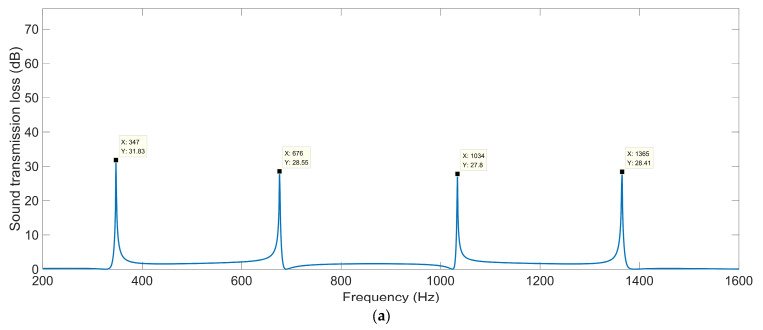
The sound-insulation performance of ultra-open acoustic-metamaterial silencer with various occupied sectional angles. (**a**) *k*_0_ = 35 deg; (**b**) *k*_0_ = 40 deg; (**c**) *k*_0_ = 45 deg; (**d**) *k*_0_ = 55 deg.

**Figure 7 materials-16-04298-f007:**
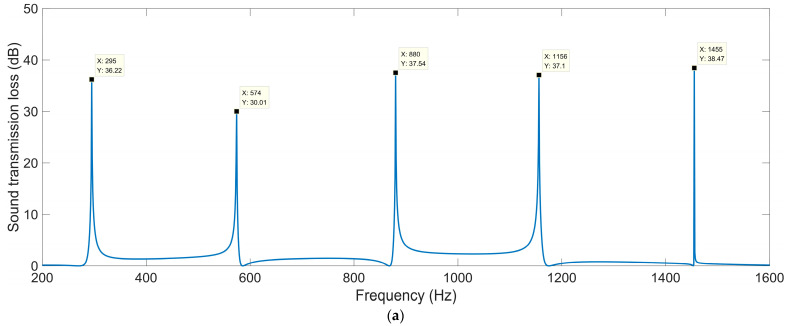
The sound-insulation performance of ultra-open acoustic-metamaterial silencer with various helix angles of each spiral chamber. (**a**) *θ* = 0.12 rad; (**b**) *θ* = 0.13 rad; (**c**) *θ* = 0.15 rad; (**d**) *θ* = 0.16 rad.

**Figure 8 materials-16-04298-f008:**
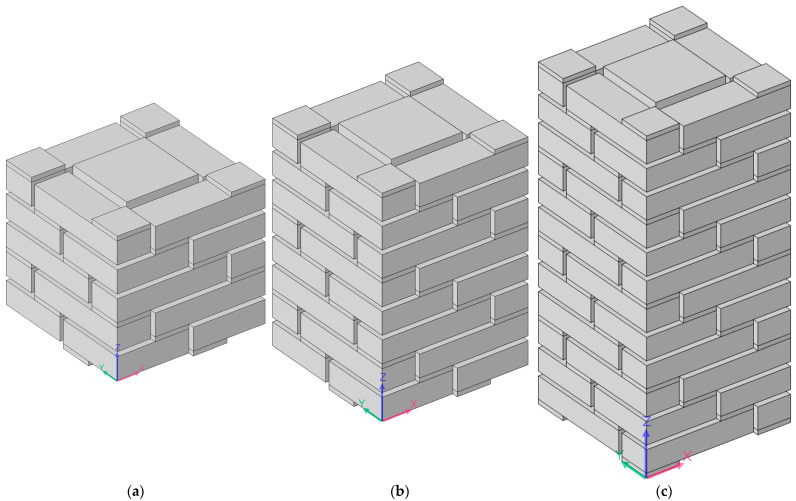
Acoustic finite element simulation models of the multiple-parallel-connection helical chambers with various numbers of layers. (**a**) *n_l_* = 5; (**b**) *n_l_* = 8; (**c**) *n_l_* = 12.

**Figure 9 materials-16-04298-f009:**
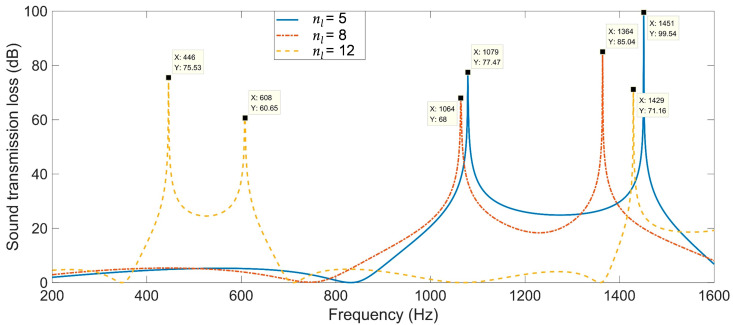
The sound-insulation performance of multiple-parallel-connection helical chambers with various numbers of layers.

**Figure 10 materials-16-04298-f010:**
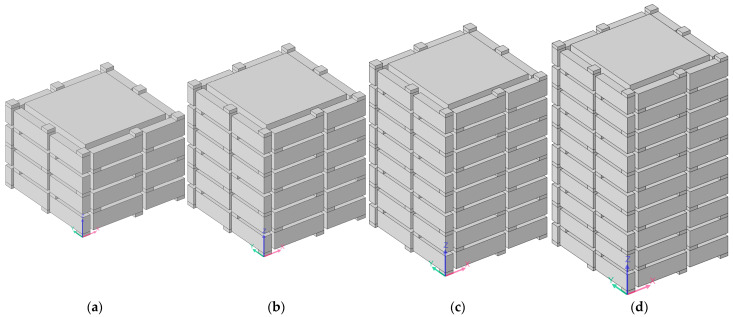
Acoustic finite element simulation model of multiple-parallel-connection folding chambers with various numbers of layers. (**a**) *n_l_* = 4; (**b**) *n_l_* = 6; (**c**) *n_l_* = 8; (**d**) *n_l_* = 10.

**Figure 11 materials-16-04298-f011:**
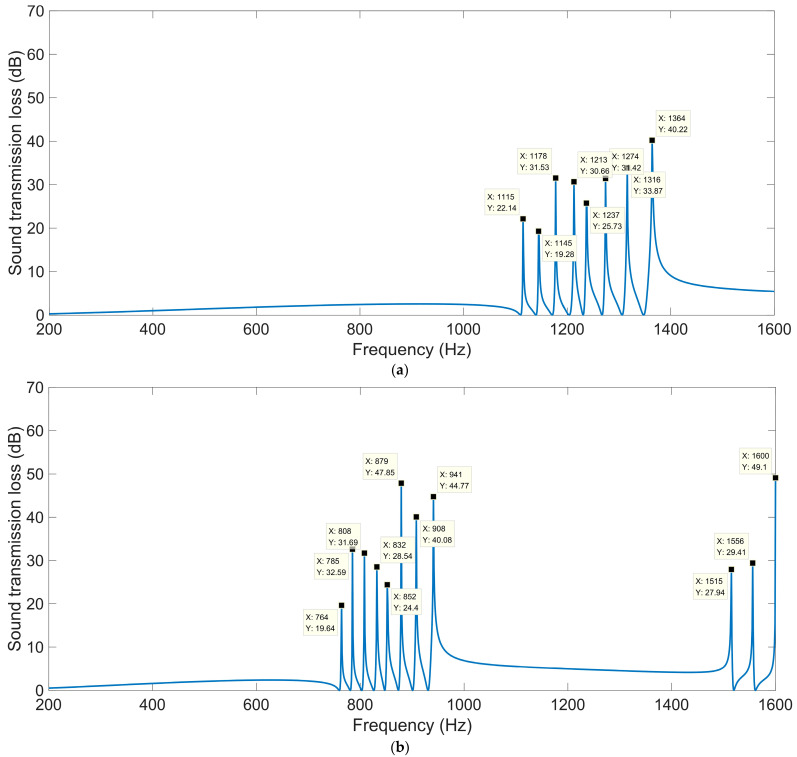
The sound-insulation performance of the multiple-parallel-connection folding chambers with various numbers of layers. (**a**) *n_l_* = 4; (**b**) *n_l_* = 6; (**c**) *n_l_* = 8; (**d**) *n_l_* = 10.

**Figure 12 materials-16-04298-f012:**
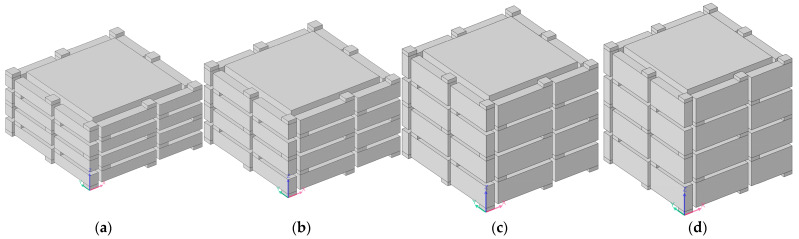
Acoustic finite element simulation model of the multiple-parallel-connection folding chambers with various thicknesses of each layer. (**a**) *L*_2_ = 6 mm; (**b**) *L*_2_ = 8 mm; (**c**) *L*_2_ = 12 mm; (**d**) *L*_2_ = 14 mm.

**Figure 13 materials-16-04298-f013:**
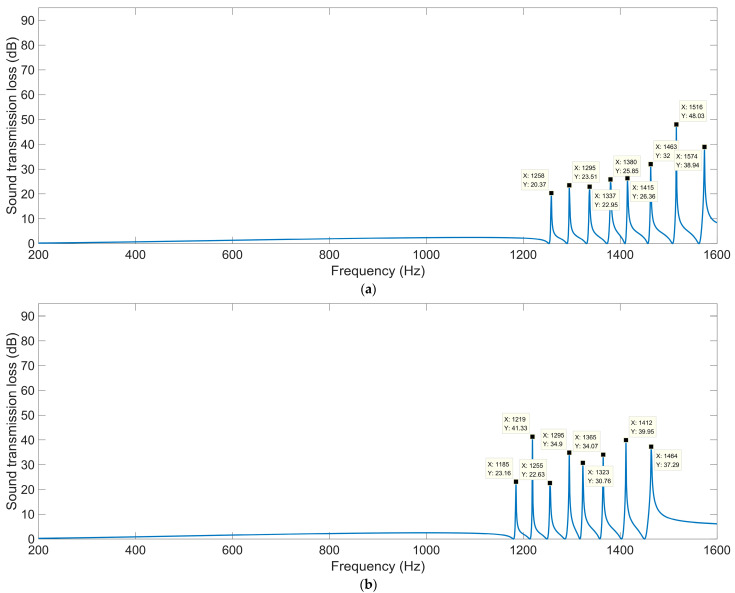
The sound-insulation performance of the multiple-parallel-connection folding chambers with various thicknesses of each layer. (**a**) *L*_2_ = 6 mm; (**b**) *L*_2_ = 8 mm; (**c**) *L*_2_ = 12 mm; (**d**) *L*_2_ = 14 mm.

**Figure 14 materials-16-04298-f014:**
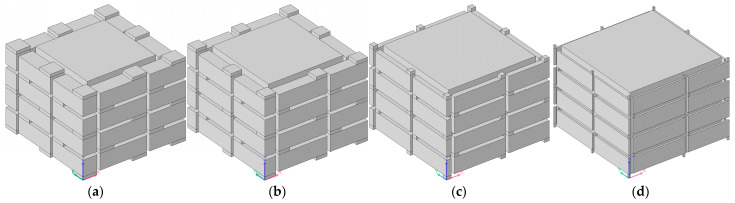
Acoustic finite element simulation model of the multiple-parallel-connection folding chambers with various inner side lengths of helical chambers. (**a**) *a*_1_ = 24 mm; (**b**) *a*_1_ = 26 mm; (**c**) *L*_2_ = *a*_1_ = 30 mm; (**d**) *a*_1_= 32 mm.

**Figure 15 materials-16-04298-f015:**
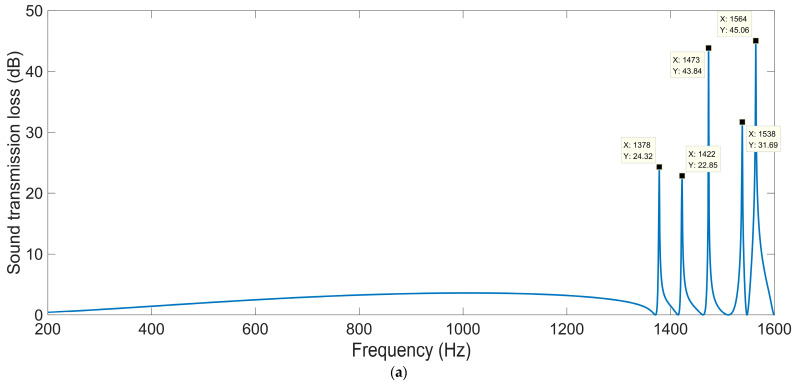
The sound-insulation performance of the multiple-parallel-connection folding chambers with various inner side lengths of helical chambers. (**a**) *a*_1_ = 24 mm; (**b**) *a*_1_ = 26 mm; (**c**) *L*_2_ = *a*_1_ = 30 mm; (**d**) *a*_1_ = 32 mm.

**Figure 16 materials-16-04298-f016:**
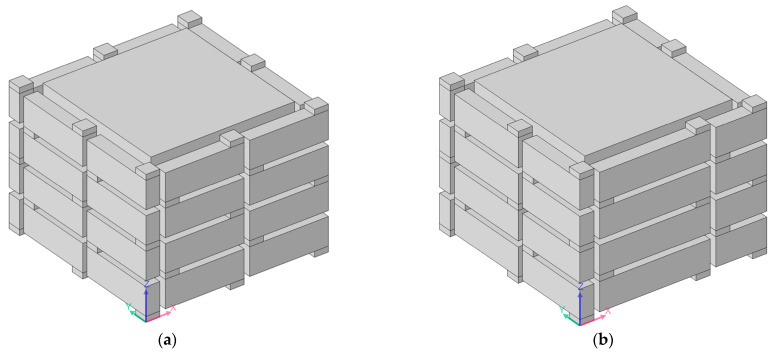
Acoustic finite element simulation model of the multiple-parallel-connection folding chambers with various intervals among various folding chambers. (**a**) *s* = 0 mm; (**b**) *s* = 3 mm.

**Figure 17 materials-16-04298-f017:**
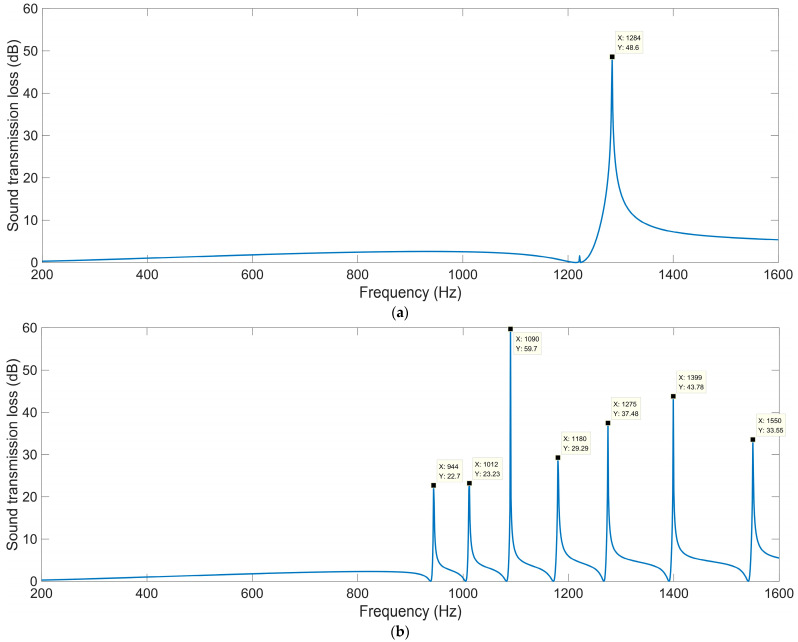
The sound-insulation performance of the multiple-parallel-connection folding chambers with various intervals among various folding chambers. (**a**) *s* = 0 mm; (**b**) *s* = 3 mm.

**Figure 18 materials-16-04298-f018:**
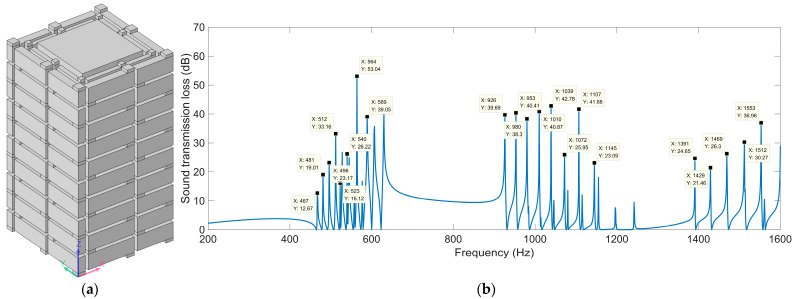
The multiple-parallel-connection folding chambers with the turns of *n_t_* = 2. (**a**) Acoustic finite element simulation model; (**b**) the sound-insulation performance.

**Figure 19 materials-16-04298-f019:**
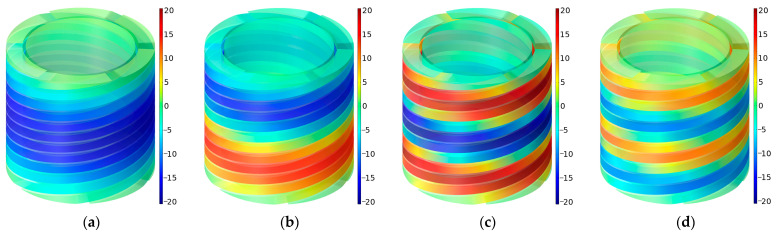
The distributions of sound pressures at the interference frequencies *f*_p_ for the ultra-open acoustic-metamaterial silencer with the default parameter values. (**a**) *f*_p_
*=* 355 Hz; (**b**) *f*_p_
*=* 684 Hz; (**c**) *f*_p_
*=* 1054 Hz; (**d**) *f*_p_
*=* 1385 Hz.

**Figure 20 materials-16-04298-f020:**
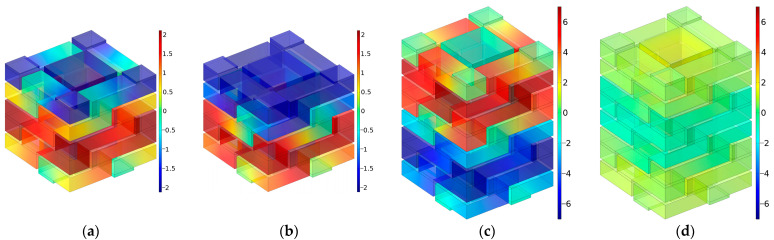
The distributions of sound pressures at the interference frequencies *f*_p_ for the multiple-parallel-connection helical chambers when *n_l_* were 5 and 8. (**a**) *n_l_* = 5 and *f*_p_
*=* 1079 Hz; (**b**) *n_l_* = 5 and *f*_p_
*=* 1451 Hz; (**c**) *n_l_* = 8 and *f*_p_
*=* 1064 Hz; (**d**) *n_l_* = 8 and *f*_p_
*=* 1364 Hz.

**Figure 21 materials-16-04298-f021:**
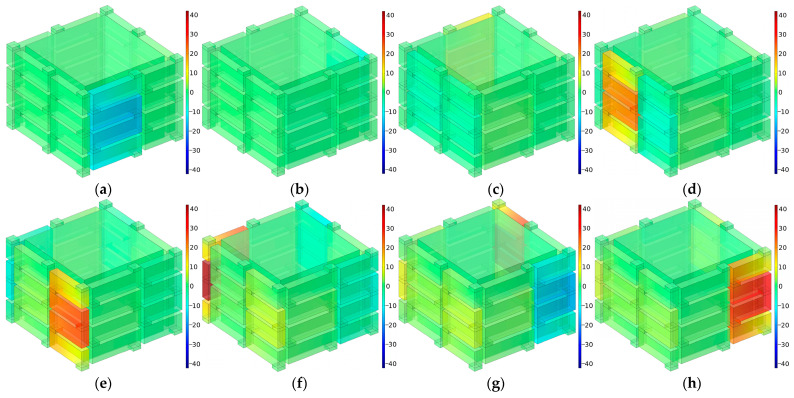
The distributions of sound pressures at the interference frequencies *f*_p_ for multiple-parallel-connection folding chambers with the default parameter values. (**a**) *f*_p_
*=* 1115 Hz; (**b**) *f*_p_
*=* 1145 Hz; (**c**) *f*_p_
*=* 1178 Hz; (**d**) *f*_p_
*=* 1213 Hz; (**e**) *f*_p_
*=* 1115 Hz; (**f**) *f*_p_
*=* 1145 Hz; (**g**) *f*_p_
*=* 1178 Hz; (**h**) *f*_p_
*=* 1213 Hz.

**Table 1 materials-16-04298-t001:** The investigated parameters and their values for investigating the sound-insulation performance of the ultra-open acoustic-metamaterial silencer.

Parameters	Symbols	Unit	Selected Values	Default Value
The total thickness	*T* _a_	mm	50 70 90 110 130	90
The open area for air passage	*μ* _open_	–	0.3 0.4 0.5 0.6 0.7	0.5
The occupied sectional angle	*k* _0_	deg	35 40 45 50 55	50
The helix angle of each spiral chamber	*θ*	rad	0.12 0.13 0.14 0.15 0.16	0.14

**Table 2 materials-16-04298-t002:** The investigated parameters and their values for studying the sound-insulation performance of the multiple-parallel-connection folding chambers.

Parameters	Symbols	Unit	Selected Values	Default Value
The numbers of layers	*n_l_*	–	4 6 8 10	4
The thickness of each layer	*L* _2_	mm	6 8 10 12 14	10
The inner side lengths of helical chamber	*a* _1_	mm	24 26 28 30 32	28
The interval among various folding chambers	*s*	mm	0 1 3	1

**Table 3 materials-16-04298-t003:** The sound-insulation property of ultra-open acoustic-metamaterial silencer with the various selected values of total thickness *T*_a_.

Various Values	1st Peak		2nd Peak		3rd Peak		4th Peak		5th Peak	
	*STL* _p_	*f* _p_	*STL* _p_	*f* _p_	*STL* _p_	*f* _p_	*STL* _p_	*f* _p_	*STL* _p_	*f* _p_
*T*_a_ = 50 mm	40.96	670	41.22	1283	–	–	–	–	–	–
*T*_a_ = 70 mm	35.94	464	44.6	892	35.95	1374	–	–	–	–
*T*_a_ = 90 mm	34.99	355	36.42	684	26.01	1054	31.25	1385	–	–
*T*_a_ = 110 mm	31.48	288	28.56	554	34.69	854	38.55	1122	12.29	1407
*T*_a_ = 130 mm	31.23	242	32.93	466	30.53	718	53.55	943	31.98	1427

**Table 4 materials-16-04298-t004:** The sound-insulation property of the ultra-open acoustic-metamaterial silencer with various selected values of the open area for air passage *μ*_open_.

Various Values	1st Peak		2nd Peak		3rd Peak		4th Peak	
	*STL* _p_	*f* _p_	*STL* _p_	*f* _p_	*STL* _p_	*f* _p_	*STL* _p_	*f* _p_
*μ*_open_ = 0.3	42.94	411	60.66	727	45.22	1176	53.25	1526
*μ*_open_ = 0.4	39.63	377	37.31	705	31.25	1106	58.05	1446
*μ*_open_ = 0.5	34.99	355	36.42	684	26.01	1054	31.25	1385
*μ*_open_ = 0.6	34.14	339	28.51	664	24.76	1012	33.86	1337
*μ*_open_ = 0.7	21.25	327	22.13	647	14.6	978	29.74	1297

**Table 5 materials-16-04298-t005:** The sound-insulation property of ultra-open acoustic-metamaterial silencer with the various selected values of occupied sectional angle *k*_0_.

Various Values	1st Peak		2nd Peak		3rd Peak		4th Peak	
	*STL* _p_	*f* _p_	*STL* _p_	*f* _p_	*STL* _p_	*f* _p_	*STL* _p_	*f* _p_
*k*_0_ = 35 deg	31.83	347	28.55	676	27.8	1034	28.41	1365
*k*_0_ = 40 deg	40.71	350	38.68	679	26.74	1040	28.94	1372
*k*_0_ = 45 deg	30.63	353	28.23	681	35.62	1047	35.4	1379
*k*_0_ = 50 deg	34.99	355	36.42	684	26.01	1054	31.25	1385
*k*_0_ = 55 deg	75.3	358	33.36	686	32.09	1060	32.01	1393

**Table 6 materials-16-04298-t006:** The sound-insulation property of ultra-open acoustic-metamaterial silencer with the various selected values of helix angle of each spiral chamber *θ*.

Various Values	1st Peak		2nd Peak		3rd Peak		4th Peak		5th Peak	
	*STL* _p_	*f* _p_	*STL* _p_	*f* _p_	*STL* _p_	*f* _p_	*STL* _p_	*f* _p_	*STL* _p_	*f* _p_
*θ* = 0.12 rad	36.22	295	30.01	574	37.54	880	37.1	1156	38.47	1455
*θ* = 0.13 rad	49.01	321	26.67	621	28.81	955	35.96	1255	–	–
*θ* = 0.14 rad	34.99	355	36.42	684	26.01	1054	31.25	1385	–	–
*θ* = 0.15 rad	33.26	373	31.31	716	33.1	1105	30.8	1455	–	–
*θ* = 0.16 rad	36.54	400	38.18	764	34.8	1179	31.71	1556	–	–

**Table 7 materials-16-04298-t007:** The sound-insulation property of multiple-parallel-connection helical chambers with various numbers of layers *n_l_*.

Various Values	1st Peak		2nd Peak		3rd Peak	
	*STL* _p_	*f* _p_	*STL* _p_	*f* _p_	*STL* _p_	*f* _p_
*n_l_* = 5	77.47	1079	99.54	1451	–	–
*n_l_* = 8	68	1064	85.04	1364	–	–
*n_l_* = 12	75.53	446	60.65	608	71.16	1429

**Table 8 materials-16-04298-t008:** The sound-insulation property of multiple-parallel-connection folding chambers with various thicknesses of each layer *L*_2_.

Various Values	1st Peak		2nd Peak		3rd Peak		4th Peak		5th Peak		6th Peak		7th Peak		8th Peak	
*STL* _p_	*f* _p_	*STL* _p_	*f* _p_	*STL* _p_	*f* _p_	*STL* _p_	*f* _p_	*STL* _p_	*f* _p_	*STL* _p_	*f* _p_	*STL* _p_	*f* _p_	*STL* _p_	*f* _p_
*L*_2_ = 6 mm	20.37	1258	23.51	1295	22.95	1337	25.85	1380	26.36	1415	32.00	1463	48.03	1516	38.94	1574
*L*_2_ = 8 mm	23.16	1185	41.33	1219	22.63	1255	34.90	1295	30.76	1323	34.07	1365	39.95	1412	37.29	1464
*L*_2_ = 10 mm	22.14	1115	19.28	1145	31.53	1178	30.66	1213	25.73	1237	31.42	1274	33.87	1316	40.22	1364
*L*_2_ = 12 mm	29.90	1051	24.30	1078	20.95	1108	41.85	1140	33.41	1160	27.33	1193	33.23	1230	53.99	1275
*L*_2_ = 14 mm	19.33	993	23.63	1017	29.53	1044	45.11	1073	33.02	1091	24.63	1120	40.47	1153	91.27	1196

## Data Availability

The data that support the findings of this study are available from the corresponding author upon reasonable request.

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
