# Peer review of "Study on Sound-Insulation Performance of an Acoustic Metamaterial of Air-Permeable Multiple-Parallel-Connection Folding Chambers by Acoustic Finite Element Simulation"

_materials, 2023, doi:10.3390/ma16124298_

Round 1

Reviewer 1 Report

Dear Authors, the paper is well structured, could be of some interest but there is a paramount lack on it: you base everything on simulation without any kind of validation. Thus, all the simulation you did are not significant. If you don't add any validation (measurmeent) of your simulation, the whole paper is not reliable and must be rejected.

Author Response

Response to reviewer’s comments

General Comment: Dear Authors, the paper is well structured, could be of some interest but there is a paramount lack on it: you base everything on simulation without any kind of validation. Thus, all the simulation you did are not significant. If you don't add any validation (measurmeent) of your simulation, the whole paper is not reliable and must be rejected.

Response:

Thank you very much for your kind review to our manuscript and constructive suggestions to our research. The acoustic finite element simulation has been proved as a reliable and effective method to investigate the sound absorption performance or the sound insulation property of the acoustic metamaterials in many literatures (some of them are listed as following). It can be judged from the literatures that the simulation result is consistent with the experimental result through selecting the appropriate parameters. Therefore, the acoustic finite element simulation method is used to investigate the sound insulation performance of the ultra–open acoustic metamaterial silencer, multiple parallel–connection helical chambers and multiple parallel–connection folding chambers.

[1] Lin Q, Lin Q, Wang Y, et al. Sound insulation performance of sandwich structure compounded with a resonant acoustic metamaterial[J]. Composite Structures, 2021, 273(1):114312.

[2] Qu T, Wang B, Min H. Lightweight Composite Partitions with High Sound Insulation in Hotel Interior Spaces: Design and Application. Buildings. 2022; 12(12):2184.

[3] Wu H, Shen Y, Liang M, Liu J, Wu J, Li Z. Performance Research and Optimization of Sound Insulation Hood of Air Compressor Unit. Applied Sciences. 2021; 11(21):10364.

[4] Du, ZZ; Chen, F; Fei, YP; Jin, JM; Li, PF; Kuang, TR; Xiao, YP; Ruan, SL; Lu, HC. High sound insulation property of prepared polypropylene/polyolefin elastomer blends by combining pressure-induced-flow processing and supercritical CO2 foaming. Composites Communications 2021, 28: 100958.

[5] Huang, YH; Lv, MY; Luo, WJ; Zhang, HL; Geng, DX. Sound insulation properties of membrane-type acoustic metamaterials with petal-like split rings. Journal of Physics D-Applied Physics 2021, 55: 045104.

Meanwhile, the simulation result of ultra–open acoustic metamaterial silencer is generally consistent with the experimental result exhibited in literature “Ghaffarivardavagh, R.; Nikolajczyk, J.; Anderson, S.; Zhang, X. Ultra–open acoustic metamaterial silencer based on Fano–like interference. Phys. Rev. B 2019, 99, 024302.”. As shown in the Figure 1, the acoustic metamaterial achieves a sound transmission loss peak at 460 Hz with the sound transmission loss of 13.6 dB. The parameters in this literature are as follows: the total thickness Ta=52 mm, the inner radius of the chamber r1 =51 mm, the outer radius of the chamber r2 = 7 cm, and the helix angle of each spiral chamber φ = 8.2â—¦. As a comparison, the built finite element simulation model of ultra–open acoustic metamaterial silencer in this research gains sound transmission loss peak at 464 Hz with sound transmission loss of 35.94 dB. The utilized parameter in our manuscript are as follows: the total thickness Ta=50 mm, the inner radius of the chamber r1 =67.9 mm (corresponding to the open area for air passage μopen=0.5), the outer radius of the chamber r2=96 mm, and the helix angle of each spiral chamber φ = 0.14 rad (≈8.0â—¦). There are two methods to derive the sound transmission loss. One is based on the sound pressure, which can be symbolled as STLp and it is utilized to show the sound insulation performance in this literature; the other is based on the sound energy, which can be symbolled as STLe and it is utilized to shown the sound insulation performance in this manuscript, and there exists the relationship of STLe=2*STLp. It can be found that the sound insulation performance in simulation in this manuscript is similar with the experimental result in the literature.

(As shown in the attachfile)

Figure 1. Experimental result in literature

(As shown in the attachfile)

Figure 2. Simulation results in this manuscript

Moreover, there are many literatures mainly studying the sound insulation performance by the numerical simulation without any experimental validations, and some of them are listed as follows. For example, a versatile and a fully coupled technique based on the finite-element-boundary element model is developed by Sahu et al. [1], and the contribution of individual component in the transmitted energy is identified by numerical simulations; It has been presented by Fan et al. [2] that theoretical analysis and finite element simulation demonstrate that the acoustic metamaterial based on membrane-coated perforated plates can effectively block acoustic waves over a wide low-frequency band regardless of incident angles; In the literature [3], the composite rubber reinforced with hollow glass microsphere is considered as a promising composite material for the noise reduction, and its sound insulation mechanism is studied based on an acoustic finite element simulation to gain the appropriate parameter with the certain constraint conditions; Kim [4] conducts the numerical simulations of the sound transmission loss of a double-panel structure  with sonic crystal comprised of distributed local resonators, and the results show that the sound insulation can be significantly improved in the low frequency range while reducing the total mass without increasing the thickness; Maurin and Kwapisz have investigated the sound transmission loss of composite barriers by means of 3D structure-acoustic finite element analysis, and the impact of different model parameters on the sound transmission loss is evaluated. Thus, we suppose that the research achievements obtained by acoustic finite element simulation are also reliable by using the suitable simulation parameters.

[1] Sahu, A; Bhattacharya, P; Niyogi, AG; Rose, M. A mobility based vibroacoustic energy transmission simulation into an enclosure through a double-wall panel. Journal of the Acoustical Society of America 2017, 141(6): EL598-EL604.

[2] Fan, L; Chen, Z; Zhang, SY; Ding, J; Li, XJ; Zhang, H. An acoustic metamaterial composed of multi-layer membrane-coated perforated plates for low-frequency sound insulation. Applied Physics Letters 2015, 106(15): 151908.

[3] Yang X, Tang S, Shen X, Peng W. Research on the Sound Insulation Performance of Composite Rubber Reinforced with Hollow Glass Microsphere Based on Acoustic Finite Element Simulation. Polymers. 2023; 15(3):611.

[4] Kim M J. Numerical study for increasement of low frequency sound insulation of double-panel structure using sonic crystals with distributed Helmholtz resonators[J]. International Journal of Modern Physics B, 2019, 33(14):1950138.

[5] Maurin A, Kwapisz L. Simplified Method for Calculating Airborne Sound Transmission Through Composite Barriers[J]. Composite Structures, 2021, 276: 114526.

[6] Giannini, D; Schevenels, M; Reynders, EPB. Rotational and multimodal local resonators for broadband sound insulation of orthotropic metamaterial plates. Journal of Sound and Vibration 2023, 547: 117453.

[7] Yu, X; Lau, SK; Cheng, L; Cui, FS. A numerical investigation on the sound insulation of ventilation windows. Applied acoustics 2017, 117: 113-121.

[8] Li S, Xu D, Wu X, Jiang R, Shi G, Zhang Z. Sound Insulation Performance of Composite Double Sandwich Panels with Periodic Arrays of Shunted Piezoelectric Patches. Materials. 2022; 15(2): 490.

[9] Ayr, U; Martellotta, F;Rospi, G. A method for the low frequency qualification of reverberation test rooms using a validated finite element model. Applied acoustics 2017, 116: 33-42.

[10] Mi, YZ; Yu, X. Sound transmission of acoustic metamaterial beams with periodic inertial amplification mechanisms. Journal of Sound and Vibration 2021, 499: 116009.

Furthermore, it is more difficult to detect the sound transmission loss for the sound insulation performance relative to detect the sound absorption coefficient for the sound absorption performance. In our previous researches, the AWA6290T standing wave tube detector has been utilized to measure the actual sound absorption coefficients of the materials or structures, as shown in the Figure 3. However, this equipment is only suitable to detect the sound transmission loss of material with the thickness smaller than 10 mm, and it cannot detect the sound insulation performance for the 3 proposed acoustic metamaterials in this study. The experimental apparatus to detect the sound transmission loss of the acoustic metamaterial is shown in the Figure 4 in the literature “Zhang H, Chen S, Liu Z, et al. Light-weight large-scale tunable metamaterial panel for low-frequency sound insulation[J]. Applied Physics Express, 2020, 13(6):067003”. It could be found that the apparatus is very complex and the required size of the sample is larger. That’s why many literatures about sound insulation performance of the acoustic metamaterial are investigated by the acoustic finite element simulation instead of the experimental measurement. Therefore, we use the acoustic finite element simulation method to investigate the 3 acoustic metamaterials in this research, and their experimental validation will be conducted in future.

(As shown in the attachfile)

Figure 3. The AWA6290T standing wave tube detector

(As shown in the attachfile)

Figure 4. The experimental apparatus to detect the sound transmission loss of acoustic metamaterial in the literature.

Reviewer 2 Report

This study proposes a novel acoustic metamaterial composed of air-permeable folding chambers to achieve a balance between sound insulation and ventilation. The material's sound insulation performance is investigated through simulation, demonstrating multiple transmission loss peaks and efficient ventilation with high sound insulation capacity. However, before further investigation of the manuscript, the authors are invited to “fully” address the comments listed below:

1-    What is the underlying principle of the Fano-like interference utilized in the acoustic finite element simulation of the proposed acoustic metamaterial?

2-    What are the advantages and limitations of the ultra-open acoustic metamaterial silencer in terms of sound transmission loss peaks and practical application value?

3-    What are the advantages of the multiple parallel-connection folding chambers over the ultra-open acoustic metamaterial silencer and multiple parallel-connection helical chambers in terms of fabrication cost and sound insulation performance?

4-    How do the total thickness (Ta), the outer and inner radius of the spiral chamber (r2 and r1), the occupied sectional angle and helix angle of each spiral chamber (k0 and θ), and the thickness of the side walls (t0) of the ultra-open acoustic metamaterial silencer contribute to its sound insulation performance, and what were the specific values and ranges explored in the investigation?

5-    Can you elaborate on the challenges associated with fabricating the ultra-open acoustic metamaterial silencer due to its complex shape, free-form curves, and three-dimensional nature, and how the proposed multiple parallel-connection helical chambers address these challenges and reduce manufacturing difficulty effectively?

6-    What were the reasons behind replacing the 4 helical chambers with the 8 folding chambers in the multiple parallel-connection folding chambers, and how did this replacement contribute to a reduction in fabrication difficulty?

7-    XYZ coordinates and scale bars are missing from many figures. 

8-    Did you do any sensitivity analysis for the numerical results obtained? Please create an “Appendix” at the end of your manuscript and provide the core “Subroutine code” of your numerical simulations (Abaqus software) in ~10-15 lines such that the readers of your paper can better understand your work.

9-    Please mention in the Introduction of your paper that apart from finite element analysis, marking insulation performance can "also" be perfromed using many other strong numerical modeling methods such as Finite Difference Method (https://doi.org/10.1016/j.apm.2019.02.023), Bezier Multi-Step Method (https://doi.org/10.1016/j.compstruct.2019.01.041), or Differential Quadrature Method (https://doi.org/10.1016/S0263-8223(97)00112-8). You can briefly introduce these methods and reference the referred papers.

10- It is recommended to introduce the elastic properties of all the structural components explained in your manuscript. You can summarize them in a table.

11- Please polish your manuscript for grammatical mistakes. 

Moderate English editing is required. 

Author Response

Response to reviewer’s comments

General Comment: This study proposes a novel acoustic metamaterial composed of air-permeable folding chambers to achieve a balance between sound insulation and ventilation. The material's sound insulation performance is investigated through simulation, demonstrating multiple transmission loss peaks and efficient ventilation with high sound insulation capacity. However, before further investigation of the manuscript, the authors are invited to “fully” address the comments listed below.

Response:

Thank you very much for your kind review to our manuscript and constructive suggestions to our research. We have revised the manuscript carefully according to your and the other reviewers’ comments. The responses to your comments are as follows.

  1. What is the underlying principle of the Fano-like interference utilized in the acoustic finite element simulation of the proposed acoustic metamaterial?

Response:

Thank you very much for your valuable question. The Fano-like interference could realize the high sound insulation performance by the interference of the sound wave penetrate the central region with those pass through surrounding spiral chambers. This presentation is added in the paragraph 4 of the ‘1. Introduction’ section in the revised manuscript.

  1. What are the advantages and limitations of the ultra-open acoustic metamaterial silencer in terms of sound transmission loss peaks and practical application value?

Response:

Thank you very much for your meaningful question. As mentioned in the paragraph 4 of the ‘1. Introduction’ section, the ultra-open acoustic metamaterial silencer could realize the high sound insulation performance by the interference of the sound wave penetrate the central region with those pass through surrounding spiral chambers, which proposed a novel acoustic metamaterial to obtain the sound insulation and ventilation simultaneously. This is pioneering by the Ghaffarivardavagh et al., which provided effective guidance for the development of more practical sound insulators. However, the shapes of ultra-open acoustic metamaterial silencer were complex, which meant the corresponding fabrications were difficult to realize and it would limit their actual applications. Meanwhile, the number of sound transmission loss peaks was quite limited generated by the ultra-open acoustic metamaterial silencer, and these were major shortcomings of this groundbreaking sound insulator. These presentations are added in the paragraph 4 of the ‘1. Introduction’ section in the revised manuscript.

  1. What are the advantages of the multiple parallel-connection folding chambers over the ultra-open acoustic metamaterial silencer and multiple parallel-connection helical chambers in terms of fabrication cost and sound insulation performance?

Response:

Thank you very much for your worthy question. As mentioned in the paragraph 3 of ‘2.3. Multiple Parallel–Connection Folding Chambers’ section, 4 helical chambers with same length for the multiple parallel–connection helical chambers in the Figure 2c were further replaced by the 8 folding chambers with various length for the multiple parallel–connection folding chambers in the Figure 3c, which was favorable to achieve additional reduction in the fabrication difficulty. It could be found from Figure 3c that the multiple parallel–connection folding chambers was periodic with the change cycle of 2 layers, and there were no twisted chambers in it. Thus, the whole acoustic metamaterial of multiple parallel–connection folding chambers could be divided into 2 kinds of front panels and 2 kinds of the corresponding cavities, which made the large–scale assembly line production possible. Moreover, the lengths of 8 folding chambers were different, which indicated that the sound wave passed the central straight hole and that threaded the 8 folding chambers realized the interference phenomenon with various frequency points in the transmission channel, and it was favorable to generate more numbers of sound transmission loss peaks. These were the major advantages of multiple parallel-connection folding chambers over the ultra-open acoustic metamaterial silencer and multiple parallel-connection helical chambers in terms of fabrication cost and sound insulation performance.

  1. How do the total thickness (Ta), the outer and inner radius of the spiral chamber (r2 and r1), the occupied sectional angle and helix angle of each spiral chamber (k0 and θ), and the thickness of the side walls (t0) of the ultra-open acoustic metamaterial silencer contribute to its sound insulation performance, and what were the specific values and ranges explored in the investigation?

Response:

Thank you very much for your valuable question. Influences of the structural parameters to the sound insulation property of ultra-open acoustic metamaterial silencer were studied and exhibited in the subsection of ‘3.1. Sound Insulation Property of Ultra–open Acoustic Metamaterial Silencer’, which included ‘3.1.1. Total Thickness Ta’, ‘3.1.2. Open Area for Air Passage μopen’, ‘3.1.3. Occupied Sectional Angle k0’, and ‘3.1.4. Helix Angle of Each Spiral Chamber θ’. The impacts of these parameters to the sound insulation performance were presented in the manuscript. Meanwhile, the outer radius of the spiral chamber r2 and the thickness of the side walls t0 were set as 96 mm and 2 mm respectively, which took the overall size and structural stability into account. The specific values and ranges selected in this study considered the practical application scenarios and actual available range simultaneously. For example, the total thickness Ta should be neither too big nor too small, because the occupied space would be too large when the Ta was too big and the sound insulation performance would be too weak when the Ta was too small. For another instance, the occupied sectional angle k0 should be no more than 60 deg (360/6), otherwise the 6 spiral chambers would interfere. Moreover, the objective of researches on impacts of these parameters would show the evolutions of sound insulation performances with the variable parameters, which could lead to the proposed acoustic metamaterials proposed in this research. These presentations are added in the paragraph 3 of the ‘2.1. Ultra–open Acoustic Metamaterial Silencer’ section in the revised manuscript.

  1. Can you elaborate on the challenges associated with fabricating the ultra-open acoustic metamaterial silencer due to its complex shape, free-form curves, and three-dimensional nature, and how the proposed multiple parallel-connection helical chambers address these challenges and reduce manufacturing difficulty effectively?

Response:

Thank you very much for your kindly suggestion. As mentioned in the paragraph 5 of the ‘2.1. Ultra–open Acoustic Metamaterial Silencer’ section, the shape of an ultra–open acoustic metamaterial silencer was complex, because there were many free–form curves and three–dimensional curves, which meant that the fabrication of actual sample was difficult to realize, especially that the complex shapes were some internal complex structures. The normal processing methods by material removal could not be utilized for the fabrication of ultra–open acoustic metamaterial silencer, because the cutting tool were unable to enter its internal with effective material machining. Although the additive manufacturing method could fabricate the acoustic metamaterials with the complex shape, it was difficult to be used in the mass production. Moreover, the complex shape of ultra–open acoustic metamaterial silencer made the mold manufacturing method was difficult to realize with the low production cost and high yield rate.

On the contrary, it could be judged from Figures 2c and 2d that there were no complex shapes for the helical chamber in the multiple parallel-connection helical chambers, and it could be divided to several layers. Each layer consisted of front panel and corresponding cavity. There were several apertures on each front panel, and there were some channels on each cavity. Thus, the whole multiple parallel–connection helical chambers could be assembled by several front panels and cavities layer by layer, which could significantly reduce the fabrication cost and improve the production efficiency relative to initial ultra–open acoustic metamaterial silencer.

  1. What were the reasons behind replacing the 4 helical chambers with the 8 folding chambers in the multiple parallel-connection folding chambers, and how did this replacement contribute to a reduction in fabrication difficulty?

Response:

Thank you very much for your meaningful question. There were two major reasons for the replacement of 4 helical chambers with 8 folding chambers in multiple parallel-connection folding chambers. Firstly, the lengths of 4 helical chambers were equal, which indicated that it could only obtain a small number of sound transmission loss peaks. Thus, in order to increase the number of sound transmission loss peaks, a multiple parallel–connection folding chambers was further developed, which had multiple folding chambers with the different lengths. Secondly, it could be found from the Figure 3c that the multiple parallel–connection folding chambers was periodic with the change cycle of 2 layers, and there were no twisted chambers in it relative to the 4 helical chambers. Thus, the whole acoustic metamaterial of the multiple parallel–connection folding chambers could be divided into 2 kinds of front panels and 2 kinds of the corresponding cavities, which made the large–scale assembly line production possible. The front panels and cavities could achieve rapid mass production by the mold manufacturing, and the expected sound insulation property would be obtained by combination of a certain number of front panels and cavities. Some additional presentations are added in paragraph 3 of ‘2.2. Multiple Parallel–Connection Helical Chambers’ section in the revised manuscript.

  1. XYZ coordinates and scale bars are missing from many figures.

Response:

Thank you very much for your valuable reminder. The XYZ coordinates and scale bars were added in the acoustic finite element simulation models in Figures 1, 2, 3, 8, 10, 12, 14, 16 and 18.

  1. Did you do any sensitivity analysis for the numerical results obtained? Please create an “Appendix” at the end of your manuscript and provide the core “Subroutine code” of your numerical simulations (Abaqus software) in ~10-15 lines such that the readers of your paper can better understand your work.

Response:

Thank you very much for your worthy suggestion. The acoustic finite element simulation was conducted in the COMSOL Multiphysics simulation software instead of the Abaqus software. Thus, there were no “Subroutine code”. The acoustic finite element simulation model could be reproduced in the COMSOL with the selected geometrical parameters and simulation parameters presented in the manuscript.

  1. Please mention in the Introduction of your paper that apart from finite element analysis, marking insulation performance can "also" be perfromed using many other strong numerical modeling methods such as Finite Difference Method (https://doi.org/10.1016/j.apm.2019.02.023), Bezier Multi-Step Method (https://doi.org/10.1016/j.compstruct.2019.01.041), or Differential Quadrature Method (https://doi.org/10.1016/S0263-8223(97)00112-8). You can briefly introduce these methods and reference the referred papers.

Response:

Thank you very much for your kindly suggestion. The recommended papers were added in the reference list in the revised manuscript, which were as follows:

[44] H. Kabir, M.M. Aghdam. A robust Bézier based solution for nonlinear vibration and post-buckling of random checkerboard graphene nano-platelets reinforced composite beams. Compos. Struct. 2019, 212: 184-198.

[45] Gu, Y.; Hua, Q.S.; Zhang, C.Z.; He, X.Q. The generalized finite difference method for long-time transient heat conduction in 3D anisotropic composite materials. Appl. Math. Model. 2019, 71: 316-330.

[46] Bert, C.W.; Malik, M. Differential quadrature: A powerful new technique for analysis of composite structures. Compos. Struct. 1997, 39: 179-189.

  1. It is recommended to introduce the elastic properties of all the structural components explained in your manuscript. You can summarize them in a table.

Response:

Thank you very much for your valuable suggestion. We did not know what do you mean “the elastic properties of all the structural components”. We suppose that there refer to the solid material of the acoustic metamaterial. In fact, the sound insulation performances of these proposed acoustic metamaterials had little relationship with the elastic properties of their solid materials, because the sound insulation was realized by the structure instead of the material. Thus, the elastic properties of the solid materials were not taken into account in this study.

  1. Please polish your manuscript for grammatical mistakes.

Response:

Thank you very much for your meaningful comment. According to your and the other reviewers’ comment, the language of whole manuscript was polished by a native speaker of English in the field of the acoustic metamaterial. Meanwhile, these modifications were highlighted in yellow in the revised manuscript.

Reviewer 3 Report

This paper proposes a novel acoustic metamaterial for sound insulation in HVAC duct systems. The proposed material was examined by numerical analyses using FEM (COMSOL Multiphysics) and extensive parametric study through the numerical results.

The paper was well-organised and writing style is good, therefore, this reviewer has only few minor comments as below:

1. In introduction, the purpose of the study can be written in more detailed manner: the sound transmission loss calculated is quite peaky and sound insulation may be limited to rather narrow band. Therefore, it is better to explain what type of noise is targeted. 

2. Page 3 (Sec. 2), LL120-121: Do 'unit sizes' mean 'element size'? It is perhaps preferrable to use 'element' instead of 'unit' here. Also, highest analysis frequency, and the relationship between its wavelength and the element size could be mentioned. Besides, element type is also an information of the interest of readers.

3. Figures 1, 2 and 3 can be better enlarged. Particularly, the characters in those figures are too small to read. Also, Tables can be written with larger font. Also, small characters are seen in the figures of calculated results. It is advisable to enlarge these characters if needed, or deleted otherwise.

4. Results: the obtained results show quite peaky characteristics, i.e., sound insulation performance is obtained in very narrow frequency range. Is it enough for the purpose of this study? Is there any mean to broaden the peaks in the characteristics?

Author Response

Response to reviewer’s comments

General Comment: This paper proposes a novel acoustic metamaterial for sound insulation in HVAC duct systems. The proposed material was examined by numerical analyses using FEM (COMSOL Multiphysics) and extensive parametric study through the numerical results. The paper was well-organised and writing style is good, therefore, this reviewer has only few minor comments as below.

Response:

Thank you very much for your kind review to our manuscript and positive assessment to our research. We have revised the whole manuscript carefully according to your and other reviewers’ comments. The responses to your comments are as follows.

  1. In introduction, the purpose of the study can be written in more detailed manner: the sound transmission loss calculated is quite peaky and sound insulation may be limited to rather narrow band. Therefore, it is better to explain what type of noise is targeted.

Response:

Thank you very much for your worthy suggestion. in most instances, the noises generated during rotational motion of the equipment are not continuous, the distributions of which are at several frequency points in the certain frequency range. Therefore, the noise control for this circumstance requires the sound insulation properties of utilized material or structure are accurate at these frequency points instead of a broadband noise reduction, which are favorable to decrease the total thickness of sound insulator, improve the sound insulation efficiency and reduce the occupied space. These presentations are added in paragraph 3 of the ‘1. Introduction’ section in the revised manuscript.

  1. Page 3 (Sec. 2), LL120-121: Do 'unit sizes' mean 'element size'? It is perhaps preferrable to use 'element' instead of 'unit' here. Also, highest analysis frequency, and the relationship between its wavelength and the element size could be mentioned. Besides, element type is also an information of the interest of readers.

Response:

Thank you very much for your valuable suggestion. The presentation of the 'unit sizes' is replaced by ‘element size’ in the revised manuscript, and the whole manuscript is checked as well. The investigated frequency range was 200-1600 Hz in this study, which took into account the normal frequency distribution of main noise generated by rotational motion of common equipment. Normally speaking, the simulation accuracy would be higher when the finite element size was smaller, and the appropriate element size was 1/50 of the minimum wavelength (1/50*340/1600*1000≈4.25 mm) for normal structures, and it should be 1/1000 of minimum wavelength (1/1000*340/1600*1000≈0.2 mm) for relative complex structures. Therefore, the grid model is shown in the Figure 1b, and the maximum and minimum element sizes were set as 5 mm and 1 mm respectively, and the utilized element type was regular tetrahedral mesh in this research. Meanwhile, in order to improve the accuracy of the investigated acoustic metamaterial, the minimum element size was set as 0.025 mm for domain of ultra–open acoustic metamaterial silencer. These presentations are added in paragraph 2 of the ‘2.1. Ultra–open Acoustic Metamaterial Silencer’ section in the revised manuscript.

  1. Figures 1, 2 and 3 can be better enlarged. Particularly, the characters in those figures are too small to read. Also, Tables can be written with larger font. Also, small characters are seen in the figures of calculated results. It is advisable to enlarge these characters if needed, or deleted otherwise.

Response:

Thank you very much for your meaningful suggestion. The Figures 1, 2 and 3 have been enlarged in the revised manuscript, which aimed to make the characters in those figures easy to read. In the original submitted manuscript, we had utilized the font size 10 for the characters in the Tables, but the editor reduced the font size from 10 to 8 in the manuscript, which made the characters in the Tables difficult to read. We have revised all the figures and Tables in the revised manuscript, which aimed to make them easy to read.

  1. Results: the obtained results show quite peaky characteristics, i.e., sound insulation performance is obtained in very narrow frequency range. Is it enough for the purpose of this study? Is there any mean to broaden the peaks in the characteristics?

Response:

Thank you very much for your helpful suggestion. As mentioned in the response to your 1st comment, the noises generated during the rotational motion of the equipment are not continuous, the distributions of which are at several frequency points in certain frequency range. Therefore, the noise control for this circumstance requires the sound insulation properties of utilized material or structure are accurate at these frequency points instead of a broadband noise reduction, which are favorable to decrease the total thickness of sound insulator, improve the sound insulation efficiency and reduce the occupied space. The peaky characteristics is the major feature the proposed acoustic metamaterials in this study, and they are enough for the purpose of this study. Based on Fano–like interference, the characteristics of these acoustic metamaterials are difficult to enlarge by themselves, and some other sound insulation materials or structures are required to form composite sound insulator for the expansion of the sound insulation characteristics, and this is the future aim of our research.

Round 2

Reviewer 1 Report

Dear authors. Thank you for your answers. If the validation is not done by you in this paper please include the discussion you sent me in the new version of the work, so as the reader can be aware of it

Author Response

Response to reviewer’s comments

General Comment: Dear authors. Thank you for your answers. If the validation is not done by you in this paper, please include the discussion you sent me in the new version of the work, so as the reader can be aware of it.

Response:

Thank you very much for your kind review to our manuscript and constructive suggestions to our research. We have added the discussion about the analysis of effectiveness and reliability of this research based on the acoustic finite element simulation in the section “3.4 Reliability Analysis” in the revised manuscript. Meanwhile, the new refereed literatures are added in the reference list as well.

Thank you again for your diligent comment.

Reviewer 2 Report

The authors addressed my comments and the manuscript can be published in the present format. 

Author Response

Response to reviewer’s comments

General Comment: The authors addressed my comments and the manuscript can be published in the present format.

Response:

Thank you very much for your kind review to our manuscript and positive evaluation to our research. Meanwhile, we have revised the manuscript carefully according to your and the other reviewers’ comments. Thank you again for your diligent comment.